# Political Dilemmas in the Making of a Sustainable City-Region: The Case of Istanbul

**Zeynep Enlil * and İclal Dinçer** 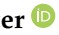

Department of City and Regional Planning, Faculty of Architecture, Yıldız Technical University, Barbaros Boulevard, Besiktas, Istanbul 34349, Turkey; diclal@yildiz.edu.tr
* Correspondence: enlil@yildiz.edu.tr; Tel.: +90-532-384-9336

**Abstract:** This article aims to explore the political dilemmas of sustainable metropolitan development marked by intense tensions between ecology and economy within the context of neoliberal urban policies over the Case of Istanbul, Turkey. It investigates the re-scaling and centralization of the state in directing the investment capital and focuses on the ways in which it reregulates and loosens the institutions to create exceptionalities in order to realize mega projects. It examines Canal Istanbul and the "New City" or the Yenişehir Project, the so-called "crazy project" imposed upon the city by the central government, which presents a crucial case demonstrating the processes of creating exceptionalities and the erosion of public norms. Empirically, drawing from the Turkish experience through an in-depth analysis of policy documents, plans and reports prepared by a variety of agents, the article demonstrates and discusses different modalities of creating exceptions to capitalize on the lucrative real estate markets through mega projects in an increasingly authoritarian neoliberal context, its ramifications on the existing norms and the oppositions it raised. The article concludes with a discussion on how the new political climate that moved away from subsidiarity, transparency and democratic participation, and became increasingly centralized, created an impasse for planning and that neither the ecology nor the economy could be protected and enhanced. Although economic development discourse is used to legitimize these mega projects, it is obvious that they lead to an ecocide.

**Keywords:** neoliberal urbanism; mega projects; institutional change; public norms; centralization; authoritarianism; ecocide; Canal Istanbul and Yenişehir Project

## 1. Introduction

Planning in complex metropolitan systems is always a contested terrain overridden by many paradoxes exacerbated by the neoliberal policies of the last four decades. Although neoliberal urbanism has variegated forms, "involving geographically uneven and path-dependent processes" [1] (p. 327), [2], it engenders a process in which relations between the state and the economy are reconstituted and state institutions become the prime agents in promoting market-based regulatory arrangements [2] (p. 102). Without doubt, neoliberal policies have spatial ramifications [3] and profound impacts on metropolitan development leading to the emergence of powerful actors with divergent stakes on urban land, formation of various growth coalitions [4], shifting power configurations, growing political pressures and authoritarian state interventions [5–7] Sustainable development of metropolitan cities is thus jeopardized by new urban policies that thrive on the urbanization of capital through the exploitation of highly speculative urban land markets and the commodification of urban space through privatization and deregulation [8]. Instrumental in this neoliberal urbanization have been mega projects, large-scale infrastructure projects and urban regeneration schemes, building high-rise offices and creating new centers. All of these projects were targeted at generating urban rent and profitable economic activities. They were part and parcel of place marketing strategies geared toward attracting loose capital in search of economic growth and competitiveness [9].

Most often, these large-scale projects were, in the Western context, realized through "exceptionality measures," which include "the freezing of conventional planning tools, by-passing statutory regulations and institutional bodies, the creation of project agencies with special or exceptional powers of intervention and decision-making, and/or a change in national or regional regulations" [10] (p. 548). The new urban policy emphasizes the primacy of project-based initiatives over regulatory plans and procedures, making "exceptionality" a major feature of neoliberal urbanism. These developments include the emergence of new policy tools, actors and institutions, all of which have significant implications for urban policymaking in general and for local democracy in particular [10] (p. 577). In this context, profit-oriented, market-based urban policies, which prioritized competitiveness and economic growth at the expense of the environment and the social good, endangered the sustainability and resilience of city-regions. Planning is thus faced with the dilemma of sustaining social and ecological concerns while at the same time maintaining economic growth that has increasingly become a priority imposed by the hegemonic power relations in a neoliberal context, and Istanbul is no exception.

Istanbul, the largest city in Europe, has been constantly under transformation pressures due to rapid urbanization since the 1950s. The last 40 years, however, have been marked by a new phase of transformation where Istanbul was envisioned as a world-class city showcasing the country's new economic policy marked by a shift from import substituting industrial economy to one seeking export oriented growth and global capital. In this newly embraced neoliberal outlook, Istanbul was envisaged as the city integrating the Turkish economy with the global markets.

As the economic engine of the country, Istanbul has always been an attraction point both for investments and people and has been under constant pressures of growth. The increasing importance of globalization, international competition and neoliberal market logic, from the 1980s onwards, intensified the pressures for urban development both at the core and the periphery of the city. Since the 1980s, various plans were made for Istanbul proposing a linear development along the shores of the Marmara Sea to the south, in order to protect the natural areas, forests and water basins located to the north. Targeting a sustainable metropolitan development, the Istanbul Metropolitan Plan of 2009 maintained the same norms. Furthermore, it proposed to fix the population of Istanbul to 16 million and limited the growth toward the north, taking into account the carrying capacity of the city and its fragile natural resources and unique ecosystem.

Regarding urban development in general and of Istanbul in particular, the State took different stances in different phases of the Turkish neoliberal experience since the 1980s [11]. Until AKP (Development and Justice Party) took the office in 2002, deregulation of the economy and refrainment from economic interventionism marked the dominant stance the previous governments adopted. In order to gain support from diverse segments of the society, a property regime that offered opportunities to various classes, ranging from lower to upper classes, was followed during the 1980s and 1990s [12].

The AKP government, which built its economic growth strategies on the urban land and property markets and the construction sector, on the other hand, took a different stance, which marked the "authoritarian turn" [13] in Turkish neoliberalization. In order to ensure the continued growth of the real estate and construction sectors, the new government increasingly became interventionist and authoritarian. It frequently made revisions in the planning legislation and vested a number of central state departments with rights and responsibilities in planning [12]. Besides, it took on an entrepreneurial approach especially in metropolitan areas such as Istanbul [14], strategically intervening in the socio-spatial restructuring of space through a variety of mechanisms. A primary mechanism that the AKP government resorted to was frequently and arbitrarily changing the norms, procedures and institutions of planning, thereby creating "exceptions" in order to pave the way for large-scale urban transformation projects. Instead of holistic and comprehensive plans that would lead the urban development, such project-based and partial interventions became the norm in urban policies. Urban land speculation and construction that

emerged in the 1980s, in due course fully, developed as the tools of capitalistic accumulation through urbanization.

What made the 2000s different from the previous neoliberal era was the regression to a gradually centralized planning system, increasing numbers of actors concerning the planning authority—creating a great confusion and uncertainty in the planning process and discipline—and, more strikingly, a tendency toward immunity from legal sanctions and control created by a new "framework of exceptions" consisting of acts, cabinet decrees and omnibus bills. Within this framework, building through "exceptions" became the norm [15], where a number of legislations passed, especially since the mid-2010s, vested the central government agencies with power to open valuable lands for development with "exceptional" building rights. The neoliberal politics involved arrangements where "the state assumes much of the risk while private sector takes most of the profits" [16] in [17] (p. 7).

Needless to say, the exceptionalities imposed by piecemeal mega projects from above were in contradiction with the fundamental principles of the 2009 Plan, which aimed to freeze the population of the city and limit its growth toward the north in order to protect the life sources of the city. These projects were part and parcel of two main goals: to enhance the position of Istanbul in the global order and to boost economic growth by stimulating the real estate market and the construction sector and exploiting the opportunities offered by the highly speculative land and property markets. These projects included the so-called "crazy projects," as they were announced to the public just before the elections of 2011, namely the Canal Istanbul, the new airport, third bridge over the Bosphorus and the "New City" (Yenişehir) for 1 million placed to the north, drastically challenging the sustainability goals of the plan, which strictly prohibited the growth toward north.

The tensions in the planning system were further exacerbated by an intense period of reregulation that witnessed the enactment of new laws and regulations, which eventually culminated in the radical restructuring of the state apparatus with the introduction of a presidential system in 2018, consolidating the authoritarian turn in Turkish politics. The new presidential system was instrumental in speeding up the processes for the execution of mega projects.

The reinforced structure of the centralized power provided by the new presidential system denotes the culmination of the consolidated link between urban space and capital accumulation, deepening the commodification of urban space and its exposure to expropriation processes in the name of enabling the capital's intervention in the cities and urban land. As Brenner and Theodore [18] (p. 352) denote, "[ . . . ] *while neoliberalism aspires to create a "utopia" of free markets liberated from all forms of state interference, it has in practice entailed a dramatic intensification of coercive, disciplinary forms of state intervention in order to impose market rule upon all aspects of social life,*" which applies precisely to what has been going on since the 2000s in Turkey. A new juridico-political framework regulating urbanization, by means of strategic alliances with multinational companies and giant investors—local and global—backs the deregulation policies of the government. A process Salet [19] conceptualizes as the "evaporation of institutions" has been predominant in the country where institutions have become increasingly "loose and volatile" and public norms as we have known melted into the air.

This article aims to explore the political dilemmas of sustainable metropolitan development marked by intense tensions between ecology and economy within the context of neoliberal urban policies over the case of Istanbul, Turkey, and examines the post-2010s in particular. It investigates the re-scaling and centralization of the state in directing the investment capital and focuses on the ways in which it reregulates and loosens the institutions to create exceptionalities in order to realize mega projects.

## 2. Materials and Methods

This article investigates Canal Istanbul and the "New City" or the Yenişehir Projects, which present a crucial case demonstrating the processes of creating exceptionalities and the erosion of public norms. Opening a canal parallel to the Bosphorus Strait was presented to the public with the justification that this would be an alternative passage to direct the international freight transport. However, the professional chambers, scientific community and non-governmental organizations have heavily criticized the project as it threatens the forest areas and water basins in the north of Istanbul. Canal Istanbul and Yenişehir Project, although not implemented yet, is an integral part of a grander scheme of infrastructure projects, including the Istanbul Airport, put into service in 2018, and Istanbul's third Bosphorus Bridge and ring roads in 2016, is the most striking example of the conflict between ecology and economy.

The research method is based on desk research and policy analysis of relevant legislation and reports of public institutions. The analyses of the technical reports prepared by the local government and professional chambers on the subject are evaluated in the article. In addition, the discussions created by the civil society in the printed, visual and social media were used to deepen our understanding of various dimensions of the subject from different perspectives.

We first frame our argument in the analyses of the institutional and legal arrangements made for the realization of the Canal Istanbul and Yenişehir Projects and contextualize it in the re-scaling of the state institutions. One of the most crucial institutional arrangements was the reorganization of the Ministry of Environment and Urbanization. The gradual expansion of the planning powers of the Ministry since 2011 by-passing local municipalities, and in this case the Istanbul Metropolitan Municipality, has been examined. Again, in order to realize the Canal Istanbul and Yenişehir Project, the processes of loosening the existing laws or amending them by defining exceptional situations were examined and discussed in terms of their effects. Foremost among these are Pasture Law, Soil Conservation and Land Use Law, Forest Law, Urban Development Law and the Law on the Transformation of Areas Under Disaster Risk.

In the second stage of our analysis, the Environmental Impact Assessment Report prepared for Canal Istanbul, which consists of 1623 pages and 37 annexes, has been examined [20] Within the scope of this article, an in-depth analysis of the EIA report in terms of planning, sustainability and management has been carried out. The criteria of analysis and data used in the report, its findings and the decisions were evaluated in relation to planning principles regarding sustainability. Another important document we used in our research is the 1:100,000 scale Metropolitan Plan Revision, concerning the area where the Canal Istanbul and Yenişehir Project are proposed. The plan documents, including a report of 235 pages, have been examined [21] and the changes the Revision Plan made in the 2009 Istanbul Metropolitan Plan, its contradictions with planning principles and public interest are evaluated.

The analysis of the opposition organized by the local government, civil society and professional chambers against the projects, on the other hand, is conveyed through the documents and public debates in the media. Among them are the comprehensive reports prepared by the Union of Chambers of Architects and Engineers of Turkey, which have been influential in raising public awareness and the formation of an informed oppositional front. A variety of efforts spent by the Istanbul Metropolitan Municipality in its struggle against this top-down project, including reports, books, publications, a workshop with the participation of a significant number of experts and stakeholders as well the website which contains crucial information about the Canal Istanbul, have been other major resources we relied on in our analysis.

The findings are discussed under three main headings. First, different modalities of making exceptions through institutional and legal arrangements are discussed from a relational perspective. Second, changing priorities and their implications for the sustainable development of Istanbul are analyzed. Third, the opposition rose against the project

among the scientific community, professional organizations, NGOs and civil society, and their struggle against the project in defense of planning principles and public interest are addressed. Thus, the research reveals how detrimental this mega project, an *exemplar par excellence* of neoliberal urban policies, could be for Istanbul and will potentially create an unprecedented ecological destruction and what kinds of exceptions were created for the realization of such a project and how established institutions and norms were destroyed.

## 3. Results

### 3.1. Different Modalities of Creating Exceptions—Piercing Istanbul's 2009 Metropolitan Plan

The 1/50,000 scale Istanbul Metropolitan Area Master Plan [22], which came into force in 1980, accepted the protection of the natural areas, agricultural areas, forests and water basins in the north of the city as the main objective of the plan, and in this context, it limited the urban development as a linear settlement along the shores of the Marmara Sea in the south (Figure 1). The 1994 and 1995 plans, following the 1980 plan, and, lastly, the 1/100,000 scale Istanbul Metropolitan Plan approved in 2009 [23] too, maintained the same norms (Figure 2). The 2009 Plan, which was first approved in 2006 but canceled for administrative reasons, adopted the same principles. It proposed to fix the population of Istanbul at 16 million and limit the growth to the north, taking into account its fragile natural resources and unique ecosystem. However, soon after the plan was approved, new institutional arrangements, approved plans and mega projects initiated by the central government began to invalidate the basic principles adopted by the 2009 Plan. It is clear that these projects aim to strengthen Istanbul's position in the global order and stimulate the economy through the construction sector by taking advantage of the opportunities offered by the highly speculative land and real estate markets [24].

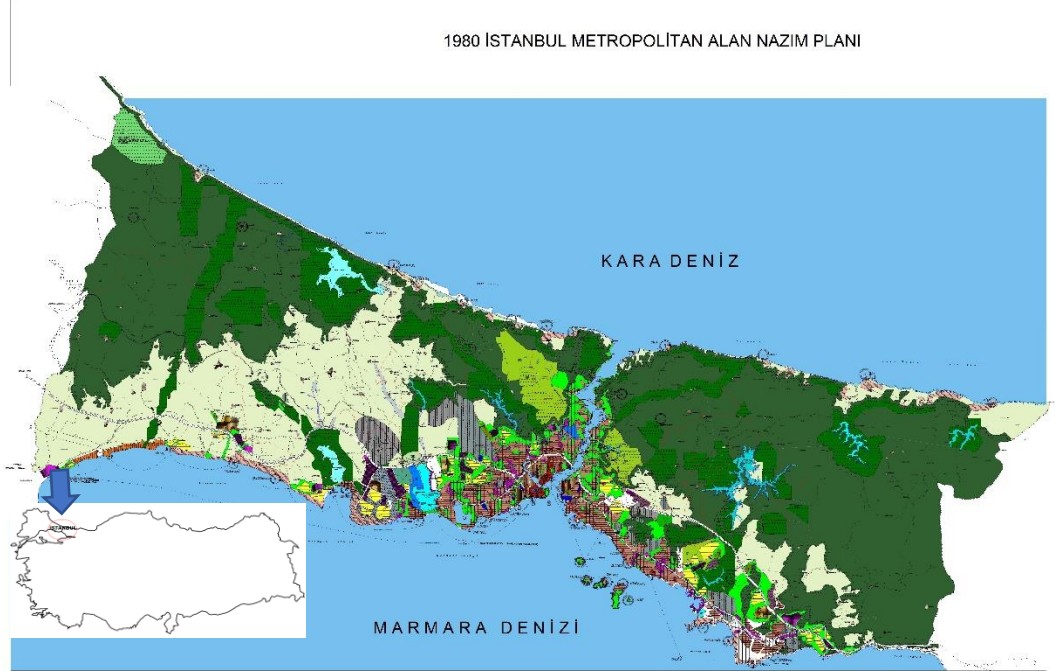

**Figure 1.** Istanbul Metropolitan Area Master Plan, 1980. https://sehirplanlama.ibb.istanbul/arsiv/ (accessed on 15 January 2022).

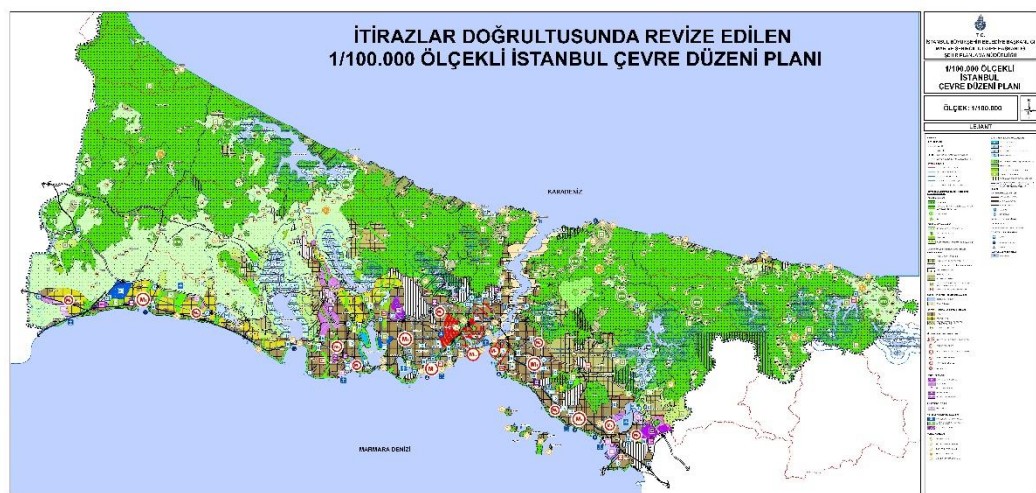

**Figure 2.** Istanbul Metropolitan Area Master Plan, 2009, https://sehirplanlama.ibb.istanbul/arsiv/ (accessed on 15 January 2022).

The main approach adopted in the 1/100,000 scale Istanbul Metropolitan Plan approved in 2006 and later in 2009 was defined as " ... handling spatial dimensions of the conflict between the economy and ecology, which is actually the product of the conflict between man and nature" [25] (p. 11). In the continuation of this definition, the report of the plan made an observation that it ends up with " ... the desire of mankind to sustain its dominance over the natural system ... results in the shortage of limited and scarce resources over time, ... [and] the occurrence of disasters ... ". Therefore, providing an environment in which these two conflicting systems could survive by nurturing each other and assuring sustainability was adopted as the basic approach of the plan. These and similar basic principles which were adopted by the 2006/2009 plans but rendered invalid by the practices of the central government are, indeed, very crucial. They serve as a warning against the ecological, economical, health and food crisis that the world is going through today.

The principles upon which the spatial macroform of Istanbul adopted in the 1/100,000 scale Istanbul Metropolitan Plan proposing a linear and incremental growth of the city in the east-west direction is defined as follows: preventing the development of the city toward the north, dividing the city into defined sub-regions within the settled area and integrating these sub-regions with their own sub-centers; connecting the urban facilities and service areas and urban sub-regional centers serving the entire metropolitan area directly to the main transportation backbone, thus establishing a healthy and functional urban organism [25] (pp. 77–78).

However, only two years after the 2009 Plan was put into effect, the central government announced the mega projects, namely the 3rd Bridge, 3rd Airport, Canal Istanbul and Yenişehir Projects within the area composed of forests, agricultural lands and water basins in the north of the city, which made the basic principles of the 1:100,000 scale Istanbul Metropolitan Plan completely null and void. Canal Istanbul Project is undoubtedly the most radical and the most devastating one among these mega projects conceived as an integrated system. This project was proclaimed on 27 April 2011 by the then Prime Minister as a project of the ruling party for the purpose of 2011 Parliamentary Elections Campaign. Recep Tayyip Erdoğan described this project, which he called the *"Crazy Project"*, *as follows:* " ... *An amazing project that I will explain.* ... *an extremely multi-purpose project.* ... *This project is an energy project, it is transportation, public works, employment, urbanism, family, housing, culture and tourism project, but above all, it is an environmental project.* ... *Today, we are rolling up our sleeves for one of the biggest projects of the century, incomparable to Panama Canal, Suez Canal or Corinth Canal in Greece.* ... " [26,27]. Being a topical issue for 10 years, this project, although drawing harsh criticisms from scientific circles, planning and preservation

specialists, professional chambers and non-governmental organizations, was prepared by the government in power for implementation in 2021 (Figure 3).

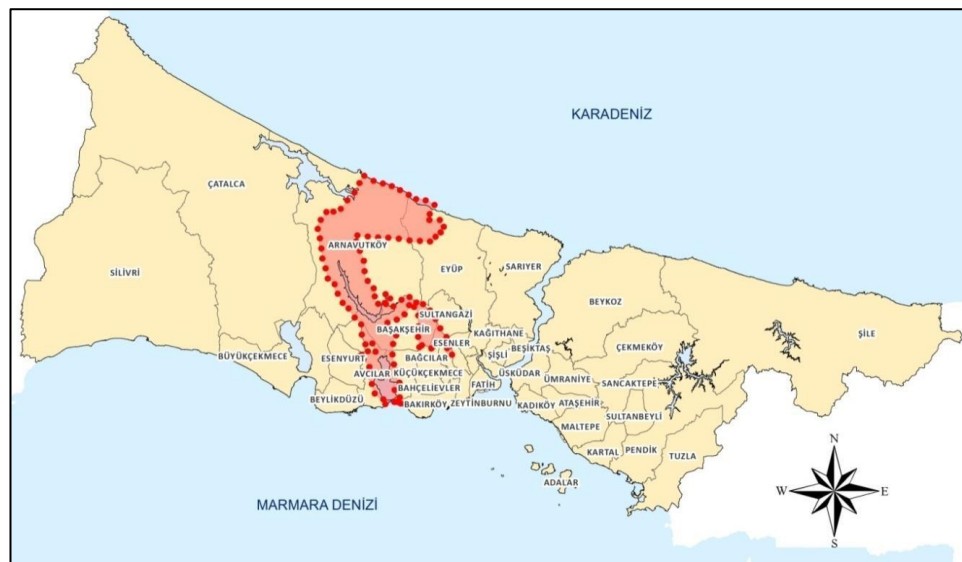

**Figure 3.** The locations of the Canal Istanbul, Yenişehir Project and İstanbul Airport in the Istanbul Metropolitan Area. The enlarged area to the north-east is where the Istanbul Airport is located. https://webdosya.csb.gov.tr/db/mpgm/icerikler/plan-deg-s-kl-g--raporu_22062020-202 00629133702.pdf (accessed on 5 January 2021).

3.1.1. Usurping of the Planning and Management Powers of the Local Government

Two major institutional arrangements marked the "authoritarian turn" in planning. The first institutional arrangement made following the announcement of Canal Istanbul to the public was to by-pass local governments and strengthen the authority of the central government: The Ministry of Environment and Urbanization (MoEU), which was established in 2011 with the reorganization of the Ministry of Public Works and Development that has been responsible for zoning and settlements for a long time, has taken its place in the system with its new structure and strengthened authorities. (The Ministry was established with the Decree Law No. 644 published in the Official Gazette dated 4 July 2011 and numbered 27,948. Shortly afterward, its powers were further strengthened by another Decree Law No. 648 published in the Official Gazette dated 17 August 2011 and numbered 28,028.)

Another important development that complemented the system following this strengthened authority was the declaration of the area where the "Canal Istanbul" and "Yenişehir Project" are located as "Development Reserve Zone for the European Side of the City of Istanbul" by the Ministry of Environment and Urbanization on 15 December 2012 (Figure 4). Shortly before this announcement, Law No. 6306 on Transformation of Areas Under Disaster Risk, or "Disaster Law," adopted on 16 May 2012, authorized the Ministry of Environment and Urbanization in the declaration and planning of reserve areas for the purpose of disaster prevention.

Declaration of the area as a "Reserve Zone" under the Disaster Law means that the planning and implementation authority over a 33,498-hectares area [28] is taken away from the authority of the Istanbul Metropolitan Municipality (IMM) and placed under the authority of the Ministry. Although the Ministry of Environment and Urbanization has similarly taken the authorities from the Istanbul Metropolitan Municipality in many parcels around Istanbul, the transfer of an area as large as this reserve area has taken place for the first time.

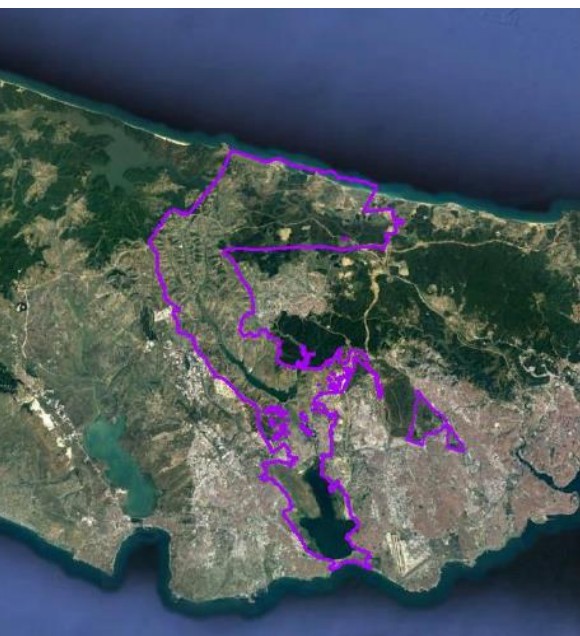

**Figure 4.** "Development Reserve Zone for European Side of the City of Istanbul". https://istanbul.csb.gov.tr/istanbul-ili-avrupa-yakasi-rezerv-yapi-alani-1-100.000-olcekli-cevre-duzeni-plani-degisikligi-duyuru-414741 (accessed on 5 January 2021).

The first project that by-passed IMM within the "Development Reserve Zone for European Side of the City of Istanbul" and was approved by the Minister of Environment and Urbanization is Istanbul Airport project, which is planned on an area of approximately 7740 hectares. All plans for this project, which was built by cutting many trees in the forest area to the north of Istanbul, as the largest airport of Europe with the largest passenger capacity, were approved ex officio by the Minister of Environment and Urbanization on 5 June 2014.

Before the Istanbul Airport project, the first project that pierced the 2009 Plan was the Third Bosphorus Bridge (Yavuz Sultan Selim Bridge) and Northern Marmara Highways project. The construction tender was held on 29 May 2012, the foundation of the bridge was laid in 2013 and the construction was completed in 2016 and put into service. Although the Ministry of Transport, Maritime Affairs and Communications approved the 3rd Bosphorus Bridge and Northern Marmara Highway route project on 4 June 2012, it could not be included in the Istanbul Metropolitan Plan. Instead, revisions were made in the plan notes of the Metropolitan Plan on 16 June 2010, which paved the way for the 3rd Bridge and similar transportation projects to be approved and entered into force through lower-scale plans [29] (p. 6). In this context, instead of directly changing the 2009 Plan, the implementation was legalized with the amendment of the "1/25,000 scale Istanbul Northern Marmara Highway Master Plan" and 17 piecemeal plans approved on 27 March 2014. It is clearly seen that the central authority, besides taking the planning and implementation authority from the local government, finds side ways and even puts it into practice without adhering to the plan approval processes in accordance with legal procedures.

3.1.2. Changing Laws and Regulations: Opening the Agricultural Lands and Pastures to Development (2016)

Following the Bridge and Airport projects in the Development Reserve Zone announced in 2012, the planning studies for the Canal Istanbul Project and the "New City" project, which is planned to be located on both sides of the canal, were initiated as of 2014. In order to prepare these plans, besides the changes made in the existing institutional arrangement, new legal arrangements have been made and norms have been changed. One of the critical changes that came into force in this context is the decision dated 30 May 2016

taken by the Ministry of Agriculture and Forestry. With this decision, 12,688 hectares of agricultural land within the "Development Reserve Zone for European Side of the City of Istanbul" has been re-zoned for non-agricultural uses. Thus, the status of agricultural zone defined in the 2009 Istanbul Metropolitan Plan, which is the main obstacle for opening agricultural areas to construction, has been removed.

These and similar practices are regulations that pierce the existing system of norms and define a new system of exceptions, in which exceptions have become the major instruments of property-led neoliberal urbanization [30]. The distribution of ground rent to certain segments of the society through piecemeal projects and instrumentalization of planning to that end, the use of planning by neoliberal ideology loosening it to create exceptions, have been predominant characteristics of the AKP period [31,32].

A similar regulation was put into effect on 14 April 2016 with an amendment to the Pasture Law No. 4342 of 1998 according to which the pasture areas are the common areas of the villages that make a living from animal husbandry. The amendment made states that "the qualifications of common goods such as pastures, summer pastures and winter quarters in the Project Area on the European Side of the City of Istanbul are removed ex officio by the Ministry of Transport, Maritime Affairs and Communications and these immovables are registered in the name of the Treasury." According to this arrangement, the status of pasture was removed from 418 properties (totaling 1343 hectares) out of a total of 440 pasturelands in the area. Therefore, an exception was created for the "Project Area on the European Side of the City of Istanbul," the authority of the Ministry of Agriculture in this area was transferred to the Ministry of Transport, making it possible for the pastures, which are common property, to be the subject of private property in the Yenişehir Project area. It is claimed that these transactions are against the procedure [33].

3.1.3. Fragmenting Planning Powers and Creating an Ambiguous Planning Environment

In parallel with the above and similar reregulations, the first call for tender for the provision of "Canal Istanbul Project Environmental Impact Assessment (EIA) Report" was made by the Ministry of Transport and Infrastructure (MoTI) General Directorate of Infrastructure Investments (GDII) on 14 July 2017. Although the EIA Report was completed on 11 December 2017, it was withdrawn a week later on 28 December 2017 with no explanation made to the public. However, the works, which were stopped due to legal deficiencies were restarted on 27 February 2018 and was completed on 23 December 2019. It is, however, useful to point out how the authorities of the Ministry of Transport and Infrastructure have also been increased since the beginning of the 2000s. In 2006, the task of " . . . planning, carrying out or having the transportation infrastructure works constructed, for connecting the two sides of the seas to each other from underneath the sea, including the build-operate-transfer model . . . " was given to the responsibility of this directorate, without the approval of the relevant municipality. In 2010, the task of preparing the plans and schedules of the rail transport systems and metros, of which the construction was committed by the Council of Ministers, too, without the approval of the relevant municipality, were included into the call of duty of the Ministry. The task of realizing the Canal Istanbul Project was given to the Ministry of Transport and Infrastructure, General Directorate of Infrastructure Investments in 2017, and this task has been confirmed and further strengthened with the Presidential Decree No. 1 on the Organization of the Presidency, published in the Official Gazette dated 10 July 2018 and radically altered the politico-institutional structure of the country. According to this, the task for realizing Canal Istanbul and similar waterway projects was included in the Ministry's call of duty. The Decree No. 1 stated the task as follows: " . . . *transportation, maritime, communication and postal affairs and services as well as carrying out works and implementation of goals identified with respect to development, implementation, having implemented, operation and having operated the Canal Istanbul and similar waterway projects, which connect the Black Sea and the Marmara Sea and which make it possible for the vessels to cruise through, in coordination with relevant institutions and organisations with a purpose to designate the national policies, strategies and goals*

. . . ". The whole aim of these restructuring is to define an exception, an extraordinary situation, and then establish its institutional and legal basis. Canal Istanbul is an exceptional project and all the arrangements made are the groundwork, the infrastructure prepared for the realization of this project. These infrastructures are also not legal in essence. The government turns the system of exceptionalities into a valid order and extends it to the whole decision-making environment.

The Ministry of Environment and Urbanization, General Directorate of Spatial Planning (GDSP), on the other hand, has started to work on 1:100,000 scale Plan Revision for the construction area (Canal Istanbul and Yenişehir Project) based on the expansion of the Ministry's planning and approval powers since 2012, as stated above. Therefore, while the Ministry of Transport and Infrastructure continued its preparations for the EIA report on Canal Istanbul, the Ministry of Environment and Urbanization began preparing the Yenişehir Project. The planning and project design work of two different Ministries in the same area led to a critical result; the EIA report and the 1/100,000 Plan Revision were completed simultaneously and the possible effects of the Yenişehir Project, a settlement of 1 million residents, were not included in any way in the EIA report. Therefore, not taking into account the impacts of such a large settlement, the EIA report has become invalid from the moment it was approved and entered into force. On the other hand, in Plan Revision for the Yenişehir Project area, the Canal Istanbul was treated as if it is just a water surface, arranged in a recreation area, without any hint of its being envisioned for the passage of large freight ships. Even this situation is enough to show how fragmented the management and planning system in which such large scale-projects are carried out. [34]. The 1/100,000 scale "Development Reserve Zone for European Side of the City of Istanbul" (Canal Istanbul and Yenişehir Project) Plan Revision was approved by the Ministry of Environment and Urbanization and put up on public display on 30 December 2019 for a period of one month.

After this public notice announcing the approval of the Plan Revision and the beginning of the objection process, objections made by a variety of actors and a large number of Istanbul residents were evaluated by the Ministry of Environment and Urbanization in accordance with the relevant legislation. The objections are discussed in detail below in Section 3.3. "Organization of the opposition against the projects." The finalization of the evaluation results could only be completed on 22 June 2020 [26]. In accordance with the zoning legislation, the upper-scale plan must be finalized before the lower-scale plans are prepared and approved. The preparation of the master plans and implementation plans for the area was envisaged in seven phases. The 1/5000 scale Master Plans and 1/1000 scale Development Plans for the 1st, 2nd and 3rd phases were again approved ex officio by the Ministry of Environment and Urbanization on 29 June 2020; only a week later the 1/100,000 Plan Revision was approved. It was announced on the website of the Ministry of Environment and Urbanization that some of the objections made during the one-month public display period regarding the 1/5000 and 1/1000 scale plans of the three phases were accepted and others were rejected, and the plan was approved ex officio on 25 March 2021.

This whole process shows that the plan approval processes at all scales are far outside the democratic approval processes of the local government that takes place in the city council. According to the zoning legislation in Turkey, the preparation and approval of master plans and implementation plans of cities, have been the authority and responsibility of local governments since 1985. However, this authority has gradually been limited, especially after the 2000s, and even totally transferred to the departments of the central government by means of exceptionality measures and institutional and legal rearrangements [28]. Therefore, in these exceptional areas, local governments are deprived of their rights to carry out democratic processes regarding making plan decisions, discussing, negotiating and approving these decisions in their respective city councils. The General Directorate of Spatial Planning of the Ministry of Environment and Urbanization is vested with the authority to approve the 1/100,000 scale Metropolitan Plan Revision for the Development Reserve Zone of the European Side of the City of Istanbul. Likewise, the Investigation

and Evaluation Commission, which was established under a ministry such as the Ministry of Transport and Infrastructure, whose task is limited to transportation, finalizes the EIA Report, while the Ministry of Environment and Urbanization carries out the approval.

This whole process shows that the planning and implementation processes related to this issue in particular and in general are carried out under the authority of the MoEU. This is proof that the authority to plan and approve, which was given to local governments in the mid-1980s, was taken back step by step in 35 years and gathered at the center again [35,36]. This process is a re-centralization process experienced throughout the country especially with the re-establishment of the MoEU in 2011 and the enactment of the Disaster Law No. 6306, by which defining privileged areas and transferring all planning powers to MoEU and Housing Development Administration (TOKI) became the rule of the game.

### 3.2. Changing Pirorities, "Evaporation" of Exisiting Norms and Implicaitions for the Sustainable Development of Istanbul

3.2.1. Environmental Impact Assessment (EIA) Report on Canal Istanbul Project

Canal Istanbul is planned to pass through the borders of four districts (Küçük Çekmece, Avcılar, Arnavutköy, Başakşehir) on the European side of Istanbul. It has been designed to cross the Küçük Çekmece Lake, which is a natural lake, penetrating in from the shores of the Marmara Sea in the south, then reaching up to the Sazlıdere Dam, which is among the important water resources of Istanbul with an annual capacity of 55 million cubic meters, has a lake area of 11.81 square kilometers and a catchment area of 165 square kilometers. From Sazlıdere Dam the Canal will continue toward north and connect to the Black Sea from near the Terkos Lake, which is also a natural lake in the Durusu Region in the north. Terkos Dam, which is a lake-dam, is the second largest dam in Istanbul with an annual capacity of 142 million cubic meters and a basin area of 619 square kilometers [37] (Figure 5). The reason for the construction of Canal Istanbul, which is envisaged as a waterway that connects the Black Sea to the Marmara Sea, which is approximately 45 km long, and 20.75 m deep and 275 m wide at its narrowest point, has been demonstrated to be easing the load of ship traffic expected to increase in the Bosphorus. It is argued that the maritime traffic in the Bosphorus, which connects the Marmara Sea and the Black Sea as a natural waterway, through which an average of 50,000 ships pass annually, will increase depending on the commercial activity in the world, and this natural waterway, which is 698 m at its narrowest point, will be insufficient. Based on the assumption that the size of the ships and especially the ships carrying dangerous/toxic fuels will threaten the Bosphorus, it was claimed that the city needs an alternative water passageway. Scientific studies carried out afterward have, however, shown that this claim is an unfounded scenario that misleads the public.

The Environmental Impact Assessment Report on Canal Istanbul Project, which was prepared by the General Directorate of Infrastructure Investments of the Ministry of Transport and Infrastructure and opened to the public's opinions and suggestions for ten days on 23 December 2019 and finally accepted on 15 January 2020 [20], consists of 1623 pages and 37 annexes. The scope of the report, which was prepared for an area of 30,019 hectares, is explained on page (p. xxi) to read " . . . examining the effects on natural life, environment, ecology and social life in detail, calculating demand forecasts by making traffic studies, preliminary and technical studies within the scope of the preparation of the construction tender documents by determining the conceptual projects, calculating the costs and determining the financing models . . . " and " . . . measurement and analysis studies to determine the current situation, marine and terrestrial flora-fauna studies, social impact assessment studies, cultural heritage impact assessment studies, surveys, drawings and engineering studies carried out on other environmental issues, . . . assessments, possible effects and measures to be taken . . . " In the EIA Report issued, technological alternatives are suggested for the measures to be taken in order to identify the "positive" or negative effects that may arise from the project and to prevent or minimize the negative effects, and it is stated that the application should be monitored and controlled. In the statement, which does not mention what the positive effects are, it is predicted that the canal will

be completed within 7 years and serve for at least 100 years, provided that the necessary maintenance is carried out. As stated in the EIA Report [20] (Chapter 3, p. 1), the Canal is a project that will affect not only the Marmara Sea but also the Black Sea and the Aegean Sea. It has been proved by the studies carried out by scientific circles that it is not possible to eliminate the negative effects of such a project using technology, and it is obvious that the maritime and terrestrial impacts will be very high in such a wide geography [38].

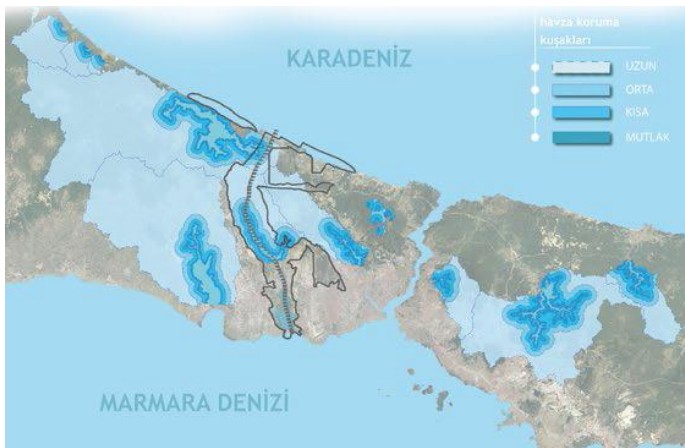

**Figure 5.** Location of Canal Istanbul and lakes, dams and water basins of İstanbul Metropoliten Area. Shades of blue shows zones with different degrees protection zones. Dark blue is where development is strictly forbidden. To the north west of the project area is the Terkos Lake, which is the major drinking water reservoir of the city. Sazlıdere Dam is located in the middle of the project area and to the south by the Marmara Sea shore is the Küçük Çekmece Lake. https://Canal.istanbul/wp-content/uploads/2020/06/CanalIstanbulCalistayi_Dijital.pdf (accessed on 15 September 2020).

In the report, it is also stated that between the years of 2011 and 2015, the General Directorate of Highways (GDH) of MoTI studied alternative routes for the Canal Istanbul Project and the route that best meets the economic and environmental conditions has been chosen among the five alternatives (Figure 6) [20] (Chapter 3, pp. 3–12). It is explained that the evaluation of these alternatives and the decision process were carried out with the support of various universities by conducting office and field studies, and the current canal route was determined as the most suitable option. However, no scientific information about the criteria with which these alternatives were evaluated and the advantages of the alternative chosen compared to other alternative routes has been shared with the public.

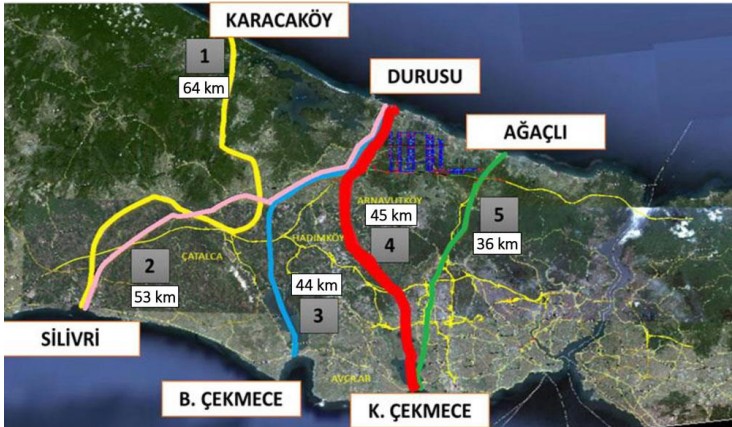

**Figure 6.** Küçük Çekmece-Sazlıdere-Durusu route no. 4 was chosen among the five alternative routes. Source: https://www.kanalistanbul.gov.tr/tr/hazirlik-sureci/alternatif-guzergah (accessed on 3 November 2021).

It is stated in the EIA Report that the area under investigation includes the canal route within the Development Reserve Zone announced by the Council of Ministers in 2012 and 2014 and the area where Yenişehir settlement is located, as well as the utilities called Marmara Port, Black Sea Port, Black Sea Land Fill Area and Black Sea Logistics Centre [20] (Chapter 3, p. 28) (Figure 7), which were the functions introduced within the scope of the 1:100,000 scale Istanbul Metropolitan Plan Revision [39]. However, there is no mention in the EIA Report of the settlement of Yenişehir, where a population of approximately one million people will be sheltered on both sides of the canal with the revision made in the Istanbul Metropolitan Plan, and the impact this population will have on the natural and cultural environment is not mentioned at all. Moreover, the report does not make any assessments about the relationship between the canal and the Yenişehir settlement. Criticisms voiced by experts about the exclusion of this residential area from the EIA assessment remain unanswered.

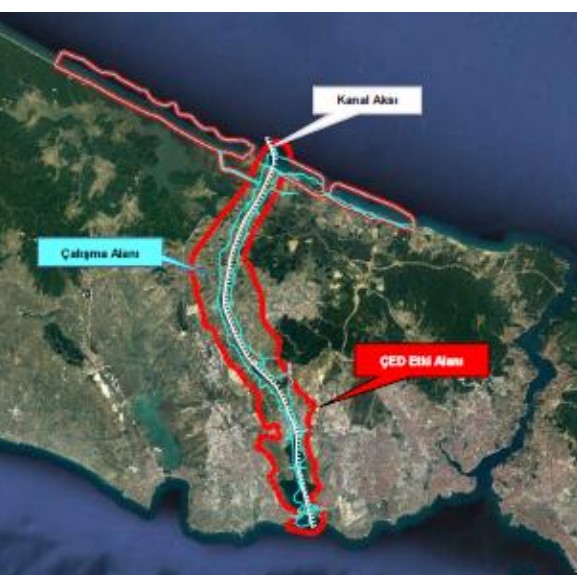

**Figure 7.** The boundaries of the area for which the EIA Report is prepared with Canal Istanbul. General Directorate of Infrastructure Investments of the Ministry of Transport and Infrastructure. https://www.kanalistanbul.gov.tr/images/uploads/icerik/KANALISTANBULNIHAICEDRAPORU.pdf, (accessed on 25 January 2021).

The diversity of the environmental impacts that should be the subject of the EIA Report and the wideness of their scope can be summarized as follows: 12 subjects are defined under the heading of project activities. These project components include a large number of projects that may affect the environment during construction and operation phases. Experts evaluating the EIA Report emphasize that the project has much more dimensions than those discussed in the report, and they criticize it for not sufficiently taking them into consideration. In the same context, Impact Factors discussed in the EIA Report is another subject that has been criticized. In the EIA Report, 18 Impact Factors are defined as physical, chemical, biological and social stress factors that will emerge as positive or negative factors at the end of the project. One of the issues that is not taken into account among these impact factors is the archaeological heritage sites in the region. Strikingly, the entrance of the Yarımburgaz Caves, which has survived for approximately 400,000 years to date, is on the Canal Istanbul route.

Impact Factor calculated as a result of the evaluation of the project activities under the aforementioned 12 subjects within the framework of the criteria of Direction + Size + Reversibility + Geographical Coverage + Duration + Frequency + Probability of Realization by matching them with 18 impact factors are classified as (1) Insignificant, (2) Low, (3) Moderate, (4) High. It is apparent that the Canal Istanbul Project has a wide range of

impacts. In this article, however, they will be discussed within a framework that is directly related to the themes of planning, sustainability and management.

3.2.2. 1/100,000 Scale Metropolitan Plan Revision (Yenişehir Project)

In this section, we examine the 1/100,000 scale 2009 Istanbul Metropolitan Plan and the 2019 Plan Revision made for the purposes of the Canal Istanbul and the Yenişehir Project, namely 1/100,000 scale Metropolitan Plan Revision, Development Reserve Zone for European Side of the city of Istanbul. This comparison is made taking into consideration the unique place of the project area within Istanbul and its cultural and natural heritage. The planning approach of the 2019 Plan Revision stated in the plan report, plan decisions and justifications are examined in this section and discussed in terms of the planning principles and sustainability.

First of all, it should be noted that, founded on the gateways of the European and Asian continents, Istanbul is a unique ancient city due to its location. Consisting of two peninsulas surrounded by the sea on three sides, Istanbul has very fragile geography and is in a uniquely rich living environment hosting the climatic characteristics of two different seas together. The region where Canal Istanbul is located is an ecological corridor formed by the Küçük Çekmece Lagoon Lake basin, Sazlıdere basin and Terkos basin merging with an old fault rupture in the north-south direction. For this reason, the region where the project is to take place contains even more unique values in Istanbul with its cultural and natural diversity [40].

As stated in the Revision Plan Report [28] (pp. 14–17), the settlements here are among the oldest settlements in the Near East and Europe, and for this reason, the region is considered to have a very important place in the history of world civilization. The Yarımburgaz Caves, located on the northern shore of Küçük Çekmece Lake, which have been used by humans and animals for about 400,000 years, will be critically affected by Canal Istanbul. It is written in Roman documents that Rhegion, the pioneer of Küçük Çekmece settlement, was founded at 18 km from the Stone of Million, a monument erected in the early 4th century AD as a zero-mile marker placed in front of Hagia Sophia. In addition, the Via Egnatia road connecting Istanbul to Rome passes through Rhegion, as well. Therefore, the Canal Istanbul and Yenişehir Project is planned in a way that threatens to destroy a very unique part of Istanbul, both in terms of its geographical values and its historical heritage [40].

The 2009 Istanbul Metropolitan Plan conceived the area where Canal Istanbul and Yenişehir Project are proposed in sharp contrast to the 2019 Plan Revision. In the 2009 Plan, new developments such as housing, tourism and education were proposed in the area to the south of the Sazlıdere Dam, which was already a partially developed area. The area to the north of the Sazlıdere Dam, composed of agricultural lands and forests, on the other hand, was protected and no new development was assigned here. (Figure 8).

Canal Istanbul and Yenişehir Project of the 2019 Plan Revision proposed a radical change for the area. In the "The Rationale for Making the Plan" section of the Revision Plan Report, after a general introduction to the 2009 Istanbul Metropolitan Plan, the rationale for the proposed plan change is explained as follows: " . . . *One of the most important components of the transportation sector is waterways and maritime. In this regard, the Rublic of Turkey wants to implement a new waterway project that will strengthen the waterways and logistics sector, integrate with major projects such as the New Istanbul Airport, alleviate the traffic load of the Bosphorus, and prevent a possible disaster due to the tankers carrying dangerous goods from here. Declared as "Canal Istanbul", this project is a project with a great impact on a wide spectrum such as the transportation sector, the logistics sector and the tourism sector*" [26] and [28] (p. 9). Although a residential area for approximately 1 million people is planned, and an employment area for 500,000 people has been arranged, there is no explanation of the rationale or justification for changing the functions organized as the Yenişehir Project in this section of the report.

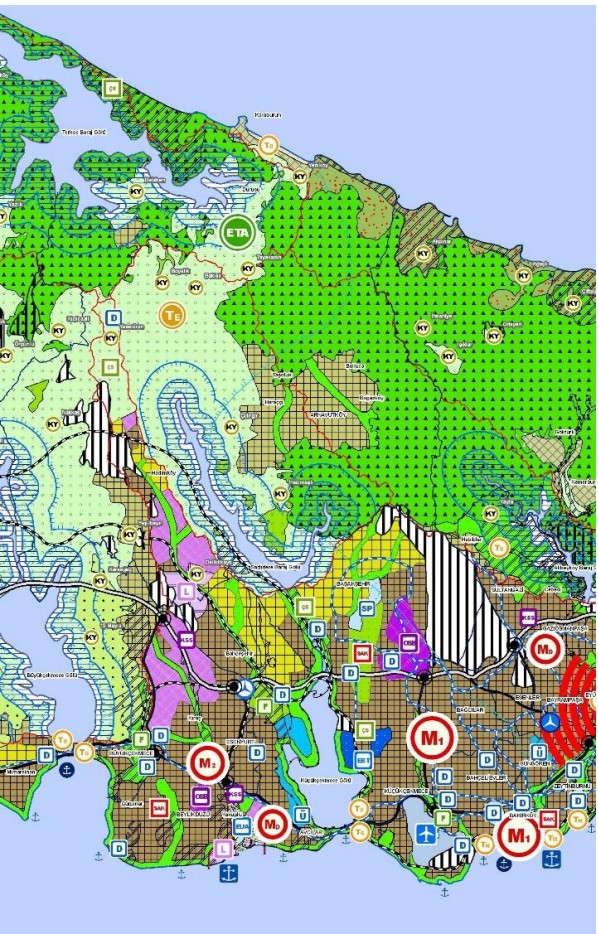

**Figure 8.** Plan decisions in the region where Canal Istanbul and Yenişehir Project are located in the 2009 Istanbul Metropolitan Plan. Note that the forests (dark green) and agricultural lands (light green) to the north of the Sazlıdere Dam are protected. Compare to Figure 10, which shows the revisions made opening all these areas to development. https://istanbul.csb.gov.tr/istanbul-ili-avrupa-yakasi-rezerv-yapi-alani-1-100.000-olcekli-cevre-duzeni-plani-degisikligi-duyuru-414741, (accessed on 5 January 2021).

In the Plan Report of the Canal Istanbul and Yenişehir Project, which was approved in 2019, ten years after the 2009 plan decision, the following statements are included while justifying these plan revision decisions: " ... *With the 2000s, the development of Istanbul was formed in an unplanned manner, sprwaling towards the north. ... The planning area ... is surrounded by unhealthy settlements and disaster-risk areas ... takes its place in the macroform of Istanbul as an axis that includes high-level investments.... it necessitates redefining the macroform of the city with disaster-risk areas, unhealty settlements, planned investment programs and projects*" [28] (pp. 43–44). As can be seen, the reason for the revision of the plan is based on "unhealthy" structures and disaster risk, and it is stated that the region will be reconstructed with new investments. However, the region offers an environment where agriculture and animal husbandry are carried out and villages, albeit partially deteriorated, still maintain their rural character (Figure 9). In addition, while the justification for the "Reserved Area" declared according to the Disaster Law should be a region to which people living in risky buildings in the earthquake-prone areas of Istanbul will be transferred, the public administration does not have any projects in this regard. On the contrary, the plots and lands in the region are rapidly changing hands in the free market and are even opened to the international real-estate market. It is claimed that 30 million square meters of land changed hands between 2011 and 2019 [38,41].

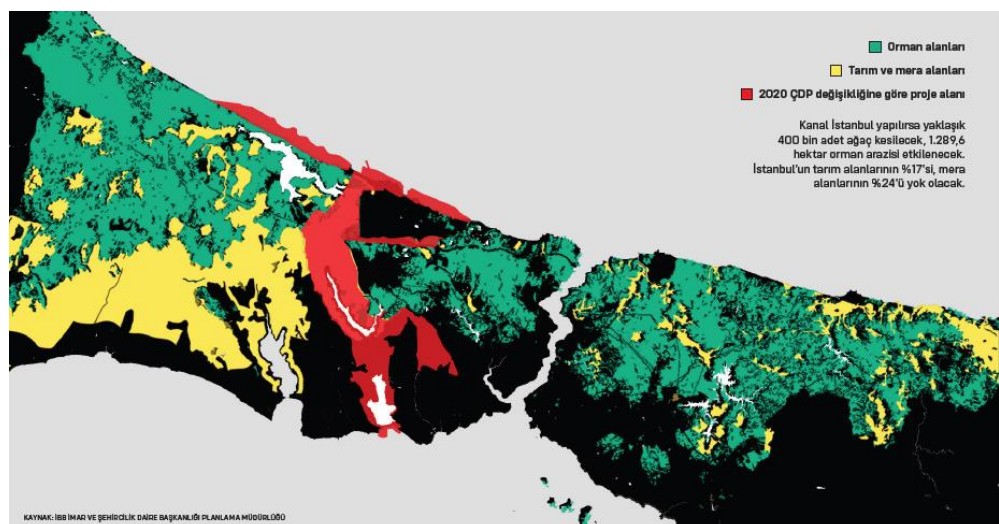

**Figure 9.** The extent of the agricultural lands (in yellow) and forest areas that will be devastated by the Canal İstanbul and Yenişehir Project, https://mekandaadalet.org/wp-content/uploads/2021/1 2/MAD_kanalistanbul_biryikimininsasi.pdf (accessed on 25 January 2022).

In the same context, it is claimed that the Canal Istanbul, Istanbul Airport, Yavuz Sultan Selim Bridge, i.e., the third bridge over the Bosphorus, and connection roads, together with the planning area, define a new axis of growth within Istanbul. It is emphasized that this axis will integrate with areas that are important in terms of ecological and natural thresholds. In the continuation of this emphasis, it is stated that the ecological values of the area should be protected and evaluated with a sustainable approach, and the development should be ensured in a controlled manner. Based on this perspective, the functional decisions brought in the plan revision made within the scope of the Canal Istanbul and Yenişehir Project are defined as four sub-regions (Figures 10 and 11): the Ecological Zone, Development Residential and Commercial Zone, Tourism and Transformation Zone and National and International Logistics Zone [28] (p. 204).

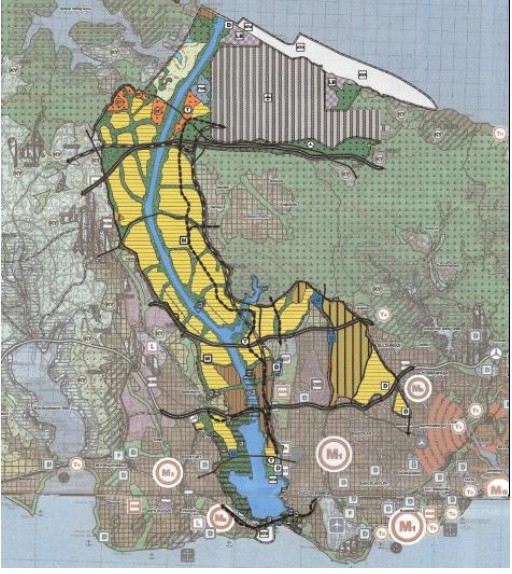

**Figure 10.** The 1:100,000 scale plan revision within the scope of Yenişehir Project, 2019. Compare with Figure 8; note the extent of the area opened to development. https://webdosya.csb.gov.tr/db/ mpgm/icerikler/cdp_f21_160dp-20191230134949.jpg (accessed on 17 September 2021).

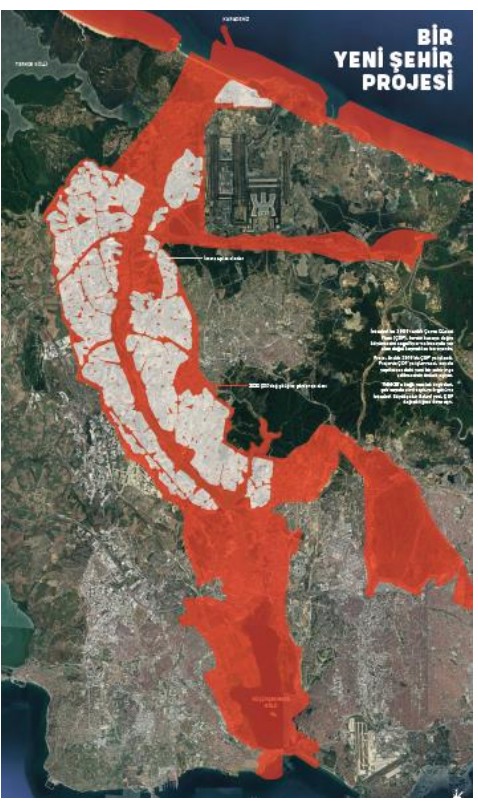

**Figure 11.** The extent of the area opened mainly to residential development by the Yenişehir Project. https://mekandaadalet.org/wp-content/uploads/2021/12/MAD_kanalistanbul_biryikimininsasi.pdf (accessed on 25 January 2022).

The first of these sub-zones is the "Ecological Zone" which is proposed within the Northern Forests of Istanbul and the second largest catchment basin in the north of the planning area. However, the discourse and decisions do not actually match since the area defined as an ecologic zone is not only very minimal in comparison to the whole planning area but also opened to development by functions such as urban facilities. The area defined as a "Residential Development and Commercial Zone" covers a wide extent of the planning area. It is claimed that this large area will be low-density and developed in accordance with the green building concept; thus, the natural character will be preserved. However, it should be remembered that these were agricultural lands that should not have been subject to urban development in any way. The third one is the "Tourism and Transformation Zone." In the Plan Revision Report, it is stated that tourism-oriented functions are brought in by making use of the tourism potential of the natural and archaeological sites in the shores of the Marmara Sea and the close vicinity of Küçük Çekmece Lake. Again, there is a mismatch between the discourse, that is, what is written in the report and the plan decisions. Although there is an attempt to promote tourism in the area, the whole project itself is a major threat to the natural and archaeological heritage jeopardizing their integrity as discussed above. A prime example of this is the case of Yarımburgaz Cave. "National and International Logistics Zone" is the Istanbul Airport and its surroundings located in the north-east of the planning area. It is defined as a logistics base at the intersection point of air, sea, land and rail transportation systems. It is obvious how negative the impacts of locating such a function in the northern forests of Istanbul will be on the environment and how it will damage the ecological values.

In addition, decisions regarding the transportation system in the area constitute one of the most detrimental decisions brought by the 2019 Plan Revision. As the scientific studies and reports cited in Section 3.3 below shows it is not possible for the plan decisions to protect the environment. On the contrary, the risks they create are enormous. One of the

most important decisions brought by the revision of the plan is the decision regarding the organization of the transportation network of the planning area. There are three major axes running across the planning area in the east-west direction. These are the D-100 Highway, TEM (Trans European Motorway) and the Northern Marmara Highway (O-7), which is under construction, connecting the European and Asian sides of the city. These three highways and other connection roads in the east-west direction cut across Canal Istanbul, which runs in the north-south direction, seven times. It is envisaged that seven crossings will be designed for road and rail systems in order to connect the east and west sides of Istanbul (Figure 12). Considering the height of the ships that will pass through the canal, the elevations of these roads and railway crossings must also be quite high. This will require the access roads on both sides of the bridges to start rising from quite a distance in order to reach the bridge height. In the light of these facts, it is inevitable that the natural and urban environment of the region where these roads will pass will have to change radically. In addition, in the event of a possible major Istanbul Earthquake, the risks to be created by connecting to the response and rescue works in the areas where the destruction occurred after the earthquake through these seven bridges are also issues that are not addressed in the EIA Report. It should not be forgotten that in case of damage to the bridges, interventions will be delayed and losses will occur [42].

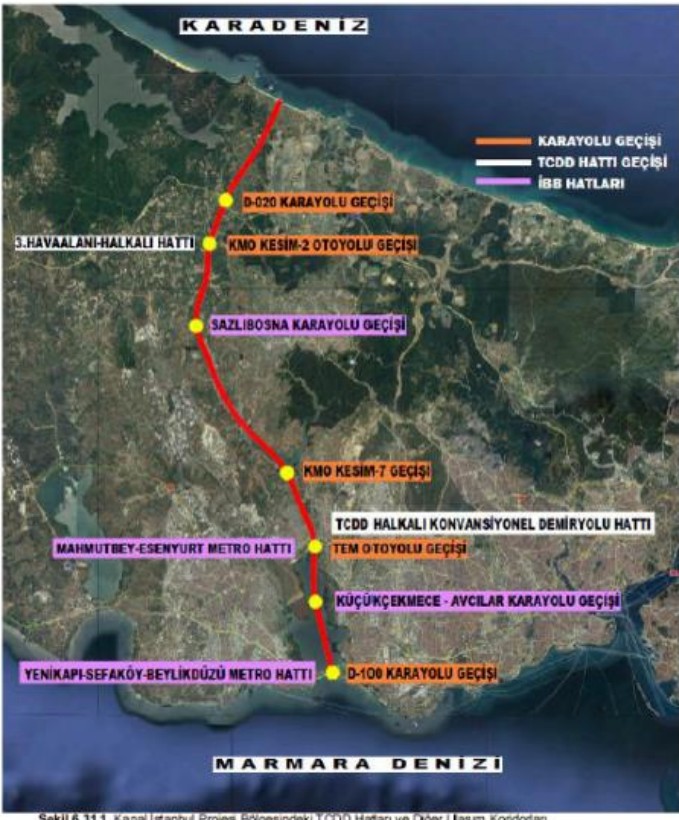

**Figure 12.** Highway and rail system crossings proposed to be located on Canal Istanbul [34] https://www.kanalistanbul.gov.tr/images/uploads/icerik/KANALISTANBULNIHAICEDRAPORU.pdf (accessed on 25 January 2021).

Canal Istanbul and Yenişehir Project, which was approved by the Ministry of Environment and Urbanization on 27 December 2019 with the 1/100.000 scale Metropolitan Plan Revision, was put into force on 22 June 2020 following the processes of public display, objection and long evaluations. One of the factors that played a role in the prolongation of the period is the simultaneous preparation of the 1/5000 scale Master Plans for the 1st, 2nd and 3rd stages of the revision plan. As mentioned earlier, these lower-scale plans were

approved a week later on 29 June 2020. Evaluation of objections to these plans also took a long time, and they came into effect 9 months later (25 March 2021). Initial investigations showed that some land-use decisions were changed in the 1/100.000 scale plan, which was approved after the examination of the objections. These revisions involved functions such as the Technology Development Zone, eco-tourism and health tourism areas, congress and exhibition area and university area, which are proposed within the Yenişehir planning area, especially in the Terkos Lake catchment area. These functions were completely removed in the 1/5000 scale plans, and almost all of the area was turned into a residential area.

As of 2019, land use status of the Development Reserve Zone for European Side of the City of Istanbul is as follows: 43.65% is composed of agricultural lands, 21.23% is the premises of the 3rd Airport, 9.85% is residential area, 8.31% is forest area, 8.29% is military area, 8.12% is lakes (Figure 13). The residential areas are mostly concentrated in the south of the area, while there are agricultural areas, forest areas and village settlements in the north apart from the airport [28] (p. 181). The completion of the Canal Istanbul and Yenişehir Project will bring about 1.2 million new inhabitants to the area. There will be a loss of 200,000 trees, 83 million sq.m. of land will be opened for development, 136 million sq.m of agricultural land and 13 million sq.m. of pasture land will be destroyed, there will be an annual loss of 33 million cubic m. of water and 17 million sq.m. of protected areas will be affected. In addition, the project will cause the creation of 3.4 million new journeys [43] (p. 21) (Figure 14).

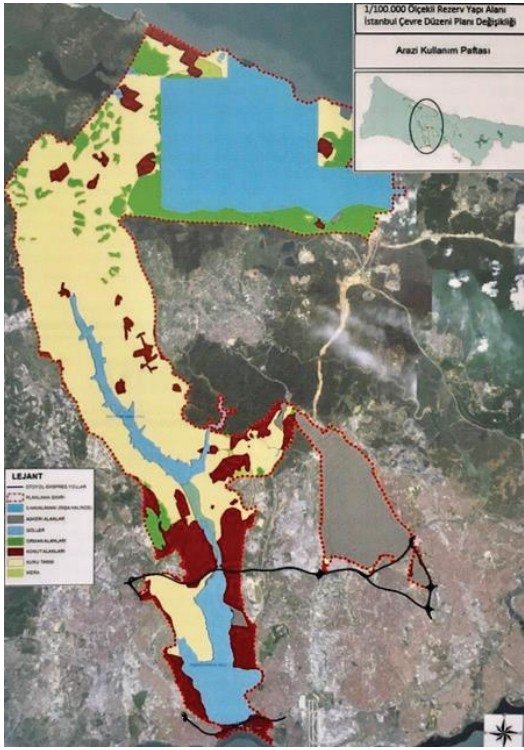

**Figure 13.** Land Use of the Reserve Zone, 2019 https://webdosya.csb.gov.tr/db/mpgm/icerikler/plan-deg-s-kl-g--raporu_22062020-20200629133702.pdf (accessed on 5 January 2021).

Although a discourse that seems to be concerned with providing sustainable development with an eye to ecological and natural thresholds, functions proposed by the 2019 Plan Revision, the allocation of land, densities and transportation system proves to be on the contrary. Mainly driven by an economic development model based on real estate and land speculation, the Plan Revision clearly shows the precedence of the economy over ecology in the increasingly centralizing policies of the AKP government.

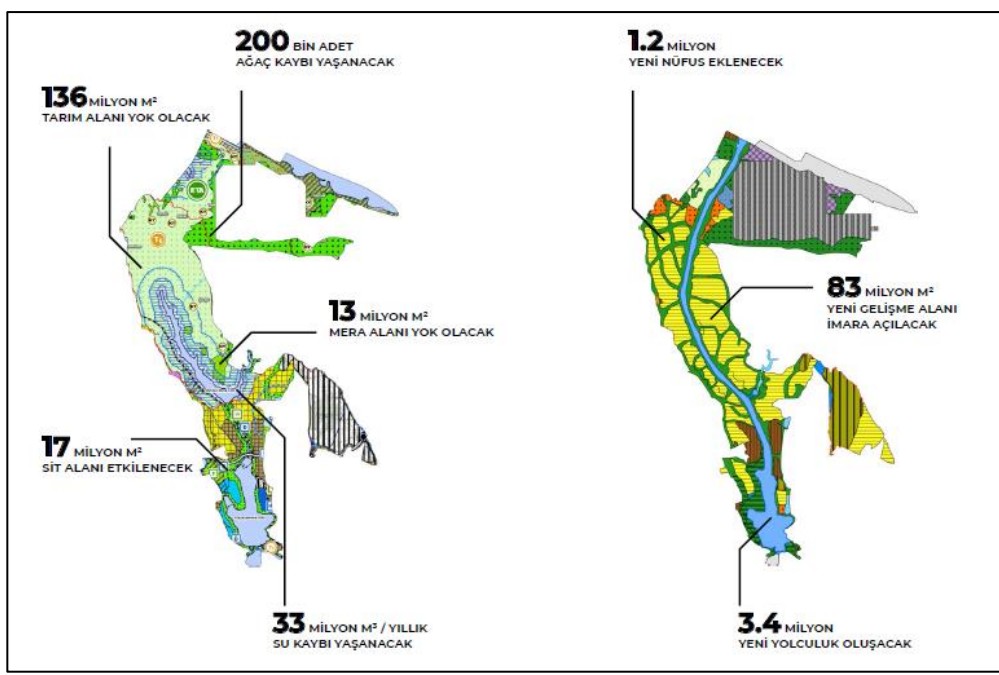

**Figure 14.** Comparison of 2009 Plan with the 2019 Plan Revision showing the natural areas that will be lost when the Canal Istanbul and Yenişehir Project is realized. https://kanal.istanbul/wp-content/uploads/2021/12/kanal_istanbul_calistay_raporu.pdf, (accessed on 15 September 2020).

*3.3. Organization of the Opposition against the Projects*

The answers to the questions "Does society want this project, or is it the corpus or is it the political power? whose needs and for what the project is being carried out?" will also illuminate despite what effects of the project will emerge. The last ten years, between 2011 and 2021, have been the years when scientists, professional chambers, non-governmental organizations and civil society, who asked these questions, struggled against these projects. The organization of the opposition against the Canal Istanbul EIA Report and the Plan Revision, or the Yenişehir Project, has been realized and continues to occur through different institutions. One of the major institutions leading this opposition front is the Union of Chambers of Architects and Engineers of Turkey. Istanbul Metropolitan Municipality (IMM), on the other hand, within the limits of its authority and responsibility, carries out objection proceedings against the project and works to inform the public about the project's inaccuracy and its devastating effects. Apart from these two main pillars of opposition, organizations such as the Turkish Anti-Erosion, Afforestation and Conservation of Natural Assets Foundation (TEMA), Northern Forests Defense and Peoples' Democratic Party (HDP) continue their opposition activities against the project. All activities and efforts help to nurture this opposition movement against this mega project and strengthen the appeal process. As a matter of fact, the submission of thousands of individual and organizational petitions to the Istanbul Directorate of Environment and Urbanization during the appeal process initiated in December 2019 for the EIA Report on Canal Istanbul and the Yenişehir Project, and even the submission of petitions from the cities other than Istanbul, shows how high the sensitivity of a significant part of the society on the subject is [44].

3.3.1. Works Carried Out by IMM

IMM has taken the realization of legal processes as a priority issue and has filed two separate lawsuits. The first lawsuit is the lawsuit filed on 13 February 2020 against the "EIA positive" decision (meaning that the proposal has no significant negative impacts on the environment) for the Canal Istanbul Project with the request of a stay of execution and cancelation, on the grounds that irreparable damages will arise if it is implemented [45]. The second lawsuit is the lawsuit filed against the Ministry of Environment and Urbanization

on 30 March 2020 for the stay of execution and cancelation of the Plan Revision concerning the Yenişehir Project [46,47]. Both litigation processes are progressing very slowly and have not yet been concluded.

On the other hand, the first event with wide participation by IMM was held on 10 January 2020 with the slogan of "Istanbul is Yours", with the "Canal Istanbul Workshop", approximately 15 days after the Canal Istanbul EIA Report and Revision Plan process started in December 2019 [43]. This approach shows how much the IMM administration attaches importance to sharing the issue with the public and struggling together. In Istanbul, where AKP mayors have ruled since 1994, CHP and Ekrem İmamoğlu, who won the local government elections on 31 March and 23 June 2019 (in the local elections held on 31 March 2019, Ekrem İmamoğlu, the candidate of the main opposition party CHP, was elected as the Istanbul Metropolitan Municipality Mayor with a margin of 13,729 votes; upon the objection of the ruling party AKP, this result was annulled by the Supreme Election Council on 6 May 2019, but Ekrem İmamoğlu was re-elected on 23 June 2019, this time with a margin of 806,014 votes), are increasing their struggle for Canal Istanbul day by day. In Turkey, where the motto "he who wins Istanbul wins Turkey" has become established during local and central election periods, the loss of this great power after 25 years by a government ruling a metropolis such as Istanbul has had a profound effect on the government. There is no end to the changes made by the central government in institutional arrangements and laws so that IMM cannot produce any services as a local government unit. These amendments made by the central government to increase its power take place through instruments such as the Decree Laws, which are the power of the President, who is the executive body, to make arrangements with the force of law, based on the temporary authority taken from the parliament, which is the legislative body. These Decree Laws at the discretion of the President lead to the "evaporation of existing institutions" [19] and the system is becoming increasingly authoritarian [48–50].

It is worth emphasizing here that Canal Istanbul, which was declared as a "crazy project" in the 2011 central elections, was never brought to the agenda by the central government and the AKP in the 2015 and 2018 central government elections nor was it used in the 2014 and even 2019 local government elections. After the 2019 elections were won by CHP candidate Ekrem İmamoğlu twice (March and June), both the 1:100,000 scale Istanbul Metropolitan Plan Revision and the EIA report were approved in December 2019. On the part of the central government, this is the rush of losing Istanbul. Losing Istanbul means the end of the power established over real estate. Hence, it means that the hegemony over the capital accumulation process is also lost. In this context, the central government frequently changes the norms curbing the planning powers of the local governments and attempts to marginalize them. In this way, it increasingly centralizes the planning powers instrumental in managing national and international capital movements in the real estate markets, using ways that contribute to further strengthening and authoritarianism of the central power [51–53].

Canal Istanbul Workshop (https://kanal.istanbul/en/, accessed on 25 January 2022), which created an effective public opinion and started with the slogan "Istanbul is Yours", was realized with the participation of more than 1500 participants, including 745 public institutions, 120 private sector, 80 professional chambers and 148 non-governmental organizations. The expressions of Mayor of Istanbul, Ekrem İmamoğlu, in his opening speech, *"We came together in this workshop to listen to scientists. As a country with very serious problems, we will see at the end of this workshop how right or how risky this project is and whether it is worth taking these risks. The attitude we show and the idea we share are never political, but absolutely vital and humane"* are of great importance. In addition, the questions raised by the Mayor *"Please stand before our children and grandchildren today and look into their eyes. Do you think they need this project, this canal? Do you think this is the smartest thing to do for their future? Do you think they expect a greener, more liveable, smoother and more civilized Istanbul from those who run this city and this country, or do they want such a risky surgery to be performed on this city?"* (https://kanal.istanbul/en/workshop-videos/, accessed on 25 January 2022) [43]. These

statements by the Mayor of Istanbul clearly show that the central government and the local government have a totally different understanding of issues. This opposition is also clearly reflected in their approach to governance. Just as the central government pursues authoritarianism by gaining strength on its own, the local one, on the contrary, tries to share power and to implement a widespread and horizontal management approach.

The report of this high-participation workshop held by IMM against the EIA Report and the Revision Plan was published in June 2020. Following this, the book titled *Multidisciplinary Evaluation of Canal Istanbul*, which was also published by IMM in July 2020, consisted of 29 articles that dealt with the subject in depth [54]. In the promotion of this book to the public, the emphasis that Canal Istanbul is a problem with risks that cannot be ignored, not only for Istanbul but also for Turkey, showed us a solidarity network that IMM is trying to establish against this great destruction of nature and the authoritarianism of the state government. IMM still continues to create public opinion, follow the lawsuits it has filed and make scientific publications. However, the rhetoric of the central government, and especially President Recep Tayyip Erdoğan, that has reached the level of "we will do it whether you want it or not" and "we will do it out of spite," still continues [55].

### 3.3.2. Studies Carried out by TMMOB—Union of Chambers of Architects and Engineers of Turkey

As soon as Istanbul Provincial Coordination Board of TMMOB started the EIA process for Canal Istanbul, it established a working group and started its work and activities as of January 2018. After the first warning was given on 28 February 2018 with the statement "We warn you! Canal Istanbul is a proposal for a geographical, ecological, economic, sociological, urban and cultural destruction and disaster", the board developed and popularized the slogan "Destruction Has No EIA" at the "for-show" EIA Report participation meeting dated 27 March 2018 held by the Ministry of Environment and Urbanization [56,57]. The prosecutor's office did not process the criminal complaint about not allowing citizens and representatives of professional chambers to attend this meeting. The people of the region who will be directly affected by the project and the headmen were not allowed to participate in the meeting, which concerns 996,000 people in 37 neighborhoods within the borders of 4 districts [58]. Two Technical Review Reports prepared by the Chamber of Environmental Engineers were published, one in May 2018 [59] and the other in June 2020 [60]. These reports are particularly significant in that they describe the institutional arrangements at the beginning of the process, newly enacted laws, lawsuits and reports prepared chronologically, together with their contents. The second report gives an elaborate and comprehensive analysis and evaluation of wide ecological impacts of Canal Istanbul and Yenişehir Project. TMMOB, at the press conference they held on 7 March 2019 [61], on the other hand, declared that Canal Istanbul was (1) an ecocide project, (2) a project that disregards planning and conservation guidelines and principles, (3) a project to increase earthquake action and the probability of devastating damage, (4) a decision that greatly harms the socio-economic life and quality of life of the local people, (5) closed to participation and (6) a project that has been put forward without making the feasibility of scientific techniques and standards as a basis. As TMMOB organized [62] non-governmental organizations and citizens to petition for the EIA Report dated 23 December 2019, it made its own institutional objection on 2 January 2020 in due time [63]. On 12 January 2020, the human chain created by Istanbulites by saying "**Either Canal or Istanbul**" was held as one of these activities, as a civil society movement in which TMMOB was also involved. TMMOB also managed and supported the objection process regarding the 1:100,000 scale Istanbul Metropolitan Plan revision approved on 27 December 2019 [63–65] and filed a lawsuit on 1 April 2020 for the stay of execution and cancelation of the plan [66]. In the first days when the COVID-19 pandemic started to affect Turkey, TMMOB shared the principle and warning of "**Budget for Health, not for the Canal**" with the public at large and the authorities, with a press release on 26 March 2020. Following the approval and public display of the 1/5000 scale Master Plan and 1/1000 scale Implementation Plan by the Ministry of Environment and

Urbanization on 29 June 2020, similar petition organizations and technical support was also provided by TMMOB. The main objection titles in these petitions were arranged as follows: In terms of contradiction to some articles of the Constitution; In terms of Pasture Law; In terms of the Law on the Transformation of Areas Under Disaster Risk; In terms of Soil Conservation and Land Use Law; In terms of Forest Law; In Terms of the Principle of Hierarchical Consistency Among Plans at Different Scales. In addition, it is requested that these plans be canceled by stating that they are clearly contrary to many principles, targets and decisions of the 2009 Istanbul Metropolitan Plan. The litigation process continues.

## 4. Discussion

The article aimed to explain the profound effects the capital, guided by neoliberal policies in Turkey since the 2000s, had on the commodification of urban land and nature. This process has emerged with the re-scaling of the state and the adoption of an increasingly authoritarian form of governmentality that pulled the planning and management powers away from the local governments toward the center. In this context, new norms, legal regulations and institutions created by defining exceptional cases in the system have directed the production of urban space. We framed our analysis of Canal Istanbul and Yenişehir Project, a mega project *alla Erdoğan*, on how the state uses "exceptionality measures" to direct investment capital and the ways in which it reregulates and loosens the institutions to realize mega projects. We empirically demonstrated and discussed above the different modalities of creating exceptionalities the AKP government in Turkey used in order to capitalize on the lucrative real estate markets through mega projects. This regime of capital accumulation through urban land speculation and built environment, which became predominant in Turkey since the 2000s, placed an emphasis on project-based initiatives over regulatory plans and procedures. Canal Istanbul and Yenişehir Project is an *exemplar par excellence* of how the relationship between the state and the economy is reconstituted over the past few decades.

### 4.1. The Impasse of Planning: Disintegration of Planning Principles, Planning Action and Public Interest

The Canal Istanbul and Yenişehir Project, as already discussed, has been a deadly blow to the 2009 Istanbul Metropolitan Plan, and the Turkish planning environment, which has already been fragmented, became increasingly ambiguous and blurred by these practices. The step-by-step acquisition of the planning powers of the local municipality by the central government since the 2000s has become a new norm imposed upon planning practice not only in the example of this project but also throughout the country. This, on the other hand, means the increasing loss of subsidiarity, governance practices, democratic rights and the loss of the concept of public interest. The scientific community, experts, IMM, professional chambers and other non-governmental organizations have heavily criticized the plan revision prepared for the Canal Istanbul and Yenişehir Project. The main axis of these criticisms is based on the fact that the plan revision made does not comply with the concept of public interest that the Constitution and various laws on the subject mandate as a guiding norm or a "normative antenna" [67], as Salet puts it. In addition, it is contrary to the norms defining planning principles that public action should be based on, which "help to cope with uncertainty in complex practices of planning" [68].

A neoliberal policy based on an increasingly authoritarian populism of AKP lacks any transparency whatsoever, and the real agenda is not made explicit to the public. It is concealed behind unfounded technical arguments justifying the construction of Canal Istanbul. As argued by Eraydın and Taşan-Kok [12] (p. 111), "in some countries, including Turkey neoliberal urban policies and practice are used to legitimize the enhancement of authoritarian governance. They argue that *authoritarian* governments use urban areas not only as a growth machine but also as grounds for a socio-political transformation project." [69]. (p. 585) For instance, it is explained in the reports that the main purpose of the Canal Istanbul Project is to alleviate the current ship traffic load of the Bosphorus.

Considering the ship characteristics that will change in the coming years, it is aimed to provide fast, safe and economical ship passages from the Black Sea to the Marmara Sea through this canal. However, maritime statistics reveal that the ship traffic passing through the Bosphorus has decreased for the last ten years, that is, from 56,606 ships to 41,103 ships and from 10,054 tankers to 8587 tankers. In addition, with the opening of the Ceyhan-Baku Pipeline, which will bring Azerbaijani oil to the Mediterranean, and the Samsun-Ceyhan Pipeline, which will transport Russian oil to the Mediterranean, the number of tankers passing through the Bosphorus every year will continue to decrease. Not considering these facts renders the rationale of the Canal Istanbul Project invalid. In addition, another argument used as the justification for Canal Istanbul is the ship accidents that took place in the Bosphorus, but statistics prove that no major accident has occurred in the last 25 years. On the other hand, no risk analysis has been made in the EIA report regarding the accidents that may occur on Canal Istanbul. Although, in case of an accident, the risk is higher for the residential areas, water basins, forest areas and Istanbul Airport that will be built around Canal Istanbul. Considering the much larger dimension of the Bosphorus, which is 698 m at its narrowest part with an average depth of 60 m, it scores much lower in terms of risks involved as compared to the Canal Istanbul, which is 275 m at its narrowest part with a depth of 20.75 m. It is clear that passing through a shallower and narrower canal compared to the Bosphorus will be much more dangerous [70]. In addition, experts state that ships will take a longer time to pass through Canal Istanbul and it will not be preferred because it is more costly and that no ship can be forced to pass through the canal as per the international agreements. Although it is very clear that the so-called main purpose of the Canal Istanbul Project diverting the ship traffic from the Bosphorus will not be fully realized, the environmental and planning problems it will create are enormous.

Considering the government's determination to put the project into practice, the question still remains how Istanbul can avoid this impasse and prevent the ecocide this "crazy project" will cause. While the debates on the real reason for insisting on this project continue, insurgencies and social movements gathered around the motto of "**Either Canal or Istanbul**" still carry the potential of "revitalizing the concept of public interest" [69]. (p. 590). The stance of the Istanbul Metropolitan Municipality is also critical here. IMM prioritizes environmental and social sustainability reflected in its vision of a "Green, Just and Creative Istanbul" and is keen on basing its opposition on data and scientific scrutiny. At the same time, with its attitude re-claiming the public good and its open-ended approach to governmentality, a window of opportunity is opened to reinstate public norms.

The local municipality, which took office in 2019, initiated a planning process to update the plan and prepare a new vision for Istanbul, targeting 2050 with a wholistic perspective emphasizing sustainability, earthquake resistance, climate resilience, socio-spatial justice and eradication of poverty through a participatory process involving a variety of stakeholders. The central government, on the other hand, with its authoritarian approach, creates a chaotic and ambiguous environment through its top-down and piecemeal decisions disrupting the balances at the metropolitan scale. In order to realize the Canal Istanbul and Yenişehir Project, the revision made in the plan concerning major decisions are related to the natural and cultural environments, agricultural lands, public works, education, employment, housing and tourism [21]. It is clear that the impact the project will have on these sectors will be very high. Although these changes require a review of all the decisions of the metropolitan plan, such a wholistic process is not carried out, but only a partial revision is made. Thus, all the balances of the 2009 plan are disturbed [71,72]. On the other hand, one of the most significant impacts of the project is a geo-political one since Canal Istanbul will divide the European part of Istanbul creating a "new island." This new island will be the heart where the CBD and all other critical sectors of Istanbul are located. In short, it is apparent that large-scale infrastructure projects such as the Canal Istanbul, Istanbul Airport and a "New City" with a population over 1 million and 500,000 jobs are extremely radical decisions that cannot be treated only as a revision to the existing plan. Rather, it requires a comprehensive approach, which should be carried out for the Metropolitan Istanbul as

a whole. Therefore, creating exceptionality at such a large-scale without conceiving how it will affect the whole is doomed to create chaos and ambiguity concerning the future of the city.

### 4.2. Changing Priorities and Threats to Ecology, Environmental Values and Cultural Assets

It became increasingly difficult for the local administrations and local actors to resist the top-down decisions imposed by this new and excessively centralized political system with a new set of political norms. Within this new political climate that moved away from subsidiarity, transparency and democratic participation, and became increasingly centralized, neither the ecology nor the economy could be protected and enhanced. Although economic development discourse is used to legitimize these mega projects, it is obvious that they lead to an ecocide.

Experts agree that the negative impacts of Canal Istanbul will be on ecosystems and living species the most. In the Canal Istanbul and Yenişehir Project area, there are ecosystems such as sea, lake, stream, swamp, dune, reed, forest, agriculture, pasture, scrub and rocky ecosystems that contain a wide variety of habitats [73]. In the review of the EIA Report on ecosystems, the impact area of the project was considered to be very narrow and was limited only to the "reserve zone," leaving the areas outside the boundaries of this zone unexamined. Even within the "reserve zone," not all the species that will be affected by the project have been defined, and measures to be taken to eliminate the negative impacts are insufficient. Besides, the ecological cost was accepted as "0" in the cost-benefit analysis made in the Canal Istanbul and Yenişehir Project. However, the real cost of the Canal will emerge if the destroyed ecosystem, habitat and living species and the services they provide are also calculated [21] (pp. 27–28)

On the other hand, TÜBİTAK (Scientific and Technological Research Council of Turkey), which provided a scientific opinion on the effects of the Canal Istanbul Project on the Marmara Sea, states that the negative consequences of the environmental impacts that will be created during the construction of the waterway are not mentioned in the EIA Report, and detailed scientific research has not been carried out on this subject. It is stated that although it is known that a large amount of material will be removed during the canal excavations, appropriate planning has not been made, the estimation model of the pollution loads and the scientific evaluation of the risks it will create on the ecosystem components have not been made by the experts. Experts agree that two different marine ecosystems such as the Black Sea and the Marmara Sea will merge unnaturally and that the Black Sea, which reaches the Marmara Sea by flowing rapidly through the canal, will completely change the whole ecosystem of the Marmara Sea, which is an inner sea with a special ecosystem [74,75].

Other natural resources and water basins will be negatively affected by the project. For instance, Küçük Çekmece Lake is one of the rare lagoon lakes. The Marmara Sea entrance and exit of the Canal Istanbul is planned at the point where this lake connects to the sea, and if it materializes, the lake will lose all its features and merge with the salty water of the sea. In this case, the lake ecosystem and the creatures living in this system will disappear. Experts strongly predict that some of the endangered species on the lists of international conservation programs in and around the lake will become extinct. Terkos Lake is the biggest drinking water dam of Istanbul. As the experts stated, the lake and lake basin are the habitats of many aquatic and semi-aquatic plant and animal species, and there are more than 10 endemic plant species under protection by international conventions. Floodplain (Longoz) forest, which is one of the rare forest species, contains a variety of species and is highly sensitive to environmental changes. These forests give the freshwater character of Terkos Lake. Thousands of birds live in the basin or use it as a haunt. The Istanbul Canal passes right next to Terkos Lake and connects to the Black Sea. Experts state that the risk of the Black Sea's water mixing with the Lake is high, that it affects the habitat of many living creatures in the Lake and that no important measures have been taken in the EIA report. The Sazlıdere Dam, which was built in 1998, is one of the important water resources

of Istanbul with its catchment area of 165 square kilometers. Sazlıdere Dam is on the route of the Canal Istanbul and will be merged into the canal and lose its function as a dam, which will decrease the water provision capacity of Istanbul. Although, it is mentioned in the EIA Report and Revision Plan Report that the lost water supply capacity will be compensated with new dams to be built in the surrounding provinces. Indeed, Istanbul Water and Sewerage Administration (ISKI) also confirms that with the disappearance of the Sazlıdere Dam, three new dams in the neighboring cities should be planned for the water supply of the European Side of Istanbul [76]. Destroying an existing water supply system only to build new ones outside Istanbul's boundaries is a most irrational solution.

In addition, it is of vital importance to protect the underground water reserves in terms of quantity and quality in the metropolitan area of Istanbul. Despite this, the opening of the Sazlıdere water catchment basin to construction and the change in the quality of the drinking water of the Terkos basin shows that the vital importance of protecting underground water reserves has not been taken into account. In addition, the existing transmission lines, wastewater collectors, network lines and treatment facilities of the city will be destroyed, put out of order or lose their function totally [76]. It is stated by experts that the effects of this project on the city's drinking water basins and underground waters, which will be irreversible when implemented, are not evaluated based on real data, and that the EIA report is very deficient and inaccurate in this respect [21] (pp. 29–34).

The supremacy of the neoliberal market logic underlying this mega project constitutes a detrimental threat not only for the natural environment but also for the cultural heritage that forms a cultural landscape of many layers. At the crossroads of two continents, Istanbul with its rich natural resources has always been a home for different people leaving traces of their civilizations. Rich layers of history are also manifest in the Canal Istanbul and Yenişehir Project Area dating back to the Paleolithic Age. The Canal Istanbul and Yenişehir Project will completely change this environment where cultural heritage and natural assets coexist, and the integrity of the cultural landscape will be shattered; components of the bio-cultural heritage in the area will also fall into ruins, will be destroyed or lose their value [21] (p. 36). Landfill areas to be constructed on the shores of the area, both on the north and the south, will destroy the cultural landscape and all traces of the history of the region. All functions, density and transportation decisions that will cause a complete change in the cultural and natural environment are completely against the conservation, planning and management principles adopted and recommended by conservation organizations around the world [21] (p. 37).

Canal Istanbul and Yenişehir Project, by connecting the two seas with different characteristics (Black Sea and Marmara Sea) with a concrete channel, disrupts the ecological balance of the region both on the sea and on land. It has, however, been proven by the scientists that the mitigation measures proposed against this threat in the Environmental Impact Assessment Reports prepared by the central government, which insisted on implementing the project, are not realistic. For nearly 80 years, to protect the forests and agricultural areas in the north, a settlement model for the development of Istanbul in the east-west direction parallel to the Marmara Sea has been adopted. It is desired to add Canal Istanbul and Yenişehir Projects to the Istanbul Airport and the Third Bosphorus Bridge, which are realized as major infrastructure projects that destroy these sources of life. Apparently, the Canal Istanbul Project is not an infrastructure project per se but a real estate project for establishing a new city (Yenişehir) with a population of over one million. In that sense, it is a clear reflection of the entrepreneurial role the state assumed, especially after 2010.

## 5. Conclusions

While the Istanbul Metropolitan Municipality continues its struggle against the project, the professional chambers are filing annulment lawsuits in order to advocate not only the right of the people of Istanbul to the city but also the protection of cultural and natural heritage areas located within the boundaries of the project, as universal values. On the other hand, this has been a learning process which brought the local government, professional

chambers and non-governmental organizations together and enhanced the culture of struggling for a common purpose in solidarity, which sparks a light of optimism for the democratization of Turkish planning practice. Regarding the current efforts toward re-institutionalization of metropolitan governance and planning in Istanbul, the Metropolitan Municipality also provides a promising ground upon which to shift the path of decades of neoliberal urbanism and reinstate public policies privileging social and environmental justice, equity and sustainability.

Shifting the path is not easy. However, there is an urgent need to search for the re-institutionalization of public norms and step away from creating exceptionalities that have become a hallmark of neoliberal urbanism, especially since the 2010s in Turkey, before it takes a deep root and is "normalized." After all, public norms are guiding principles that help keep public action on track in the context of uncertainty and align them with the common good. The future agenda of planning in Turkey, as elsewhere, should focus on restoring planning institutions that have eroded due to the ambiguity and uncertainty caused by arbitrarily changing public norms under neoliberal regimes. In terms of sustainability, what is urgently needed is a regenerative outlook that gives due priority to the common good and ecology over economy. The role of planners in steering public action must be guided by innovative practices and local struggles.

**Author Contributions:** Conceptualization, Z.E. and İ.D.; methodology, Z.E. and İ.D.; formal analysis, Z.E. and İ.D.; investigation, Z.E. and İ.D.; resources, Z.E. and İ.D.; data curation, İ.D.; writing—original draft preparation, Z.E. and İ.D.; writing—review and editing, Z.E. and İ.D.; visualization, Z.E. and İ.D. All authors have read and agreed to the published version of the manuscript.

**Funding:** This study did not receive external funding.

**Data Availability Statement:** Data are contained within the article or referenced.

**Conflicts of Interest:** The authors declare no conflict of interest.

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
