# Peer review of "Political Dilemmas in the Making of a Sustainable City-Region: The Case of Istanbul"

_sustainability, doi:10.3390/su14063299_

Round 1

Reviewer 1 Report

The authors presented the transformation of Istanbul from a sustainably planned city to capital-oriented neoliberal urban development. It looks like a policy review paper. They have discussed how the development plan of the late twentieth century got politically influenced and turned into a capitalist economy-based development in the twenty-first century at the cost of environmental sustainability.

Though the background given is in detail, explaining the actual plan (1980) of Istanbul with a map would still be better. The changes brought in the master plan were also not shown using a map. The location map of Istanbul with its boundaries is essential to understand its geographical setting. The map must show the location of new projects for a clear understanding. The impact of such transformation due to the so-called “crazy project” and the quantification of loss of natural habitats/ecological sustainability were unclear.

Since the Government is strong in implementing its plan, the alternatives to bring back balance in nature were also not discussed. In a nutshell, the paper brought out the conflict between economy and ecology in the developmental process that the people of Istanbul experience against the political will.

If the discussions are short and sharp, the paper will attract more readers and impact the ongoing development processes in Istanbul.

Repeating the same sentences may be avoided, and more graphics will enhance the succinctness.

Author Response

Response to Reviewer 1 Comments

Dear Reviewer,

Thank you so much for your constructive criticism. We improved our manuscript taking your suggestions into consideration. Below are the revisions made in response to your comments.

Once again, thank you very much for helping us to improve the manuscript. We hope that we have been able to respond to your suggestions adequately.

Best regards,

---------------------------------------------------------------------------------

Point 1: Is the content succinctly described and contextualized with respect to the previous and present theoretical background and empirical research (if applicable) on the topic?

Response 1: We tried to strengthen the contextualization of our study and added a few more references to scholarly work related to the subject. All our additions are in YELLOW

Point 2: Are the research design, questions, hypotheses, and methods clearly stated?

Response 2: We revised the abstract and added new paragraphs in Section 1. (Introduction) and in Section 2. (Material and Methods) in order to clarify our research questions, and the material we used in our analysis. Our additions to the text are marked in YELLOW.

Point 3: Are the arguments and discussion of findings coherent, balanced and compelling?

Response 3: We eliminated repetitions, shortened where appropriate, and added new phrases (marked in YELLOW) to enhance the clarity and coherence of arguments and discussions. We also did English language editing to improve the text

Point 4: Though the background given is in detail, explaining the actual plan (1980) of Istanbul with a map would still be better. The changes brought in the master plan were also not shown using a map. The location map of Istanbul with its boundaries is essential to understand its geographical setting. The map must show the location of new projects for a clear understanding. The impact of such transformation due to the so-called “crazy project” and the quantification of loss of natural habitats/ecological sustainability were unclear.

Response 4: We added both the 1980 Plan (Figure 1) and 2009 Plan (Figure 2), which shows the continuity of the planning approach among the two plans in directing the growth of the city in the east-west direction and limiting growth towards the north. Figure 1 also shows the location of Istanbul within Turkey. Figure 3 shows the extent of the project area, which includes the Canal Istanbul and Yenişehir Project as well as the Istanbul Airport. Also, Figures 4 and 5 show the project area and help to understand some of the existing features of the area where the project is located. Figure 7 shows the proposed route of the canal. Figure 9, again, shows the vast extent of the project area and the surrounding agricultural lands and forests. Figure 8 shows the portion of the 2009 plan concerning the project area, which shows clearly the area to the north of the Sazlıdere Dam is protected. Figure 10 shows the revision made where the Sazlıdere dam no longer exists and all the protected areas are opened to residential development shown in yellow.

The impact of the 2019 Plan Revision on the 2009 Plan is also shown comparatively in Figure 14. The impact of this transformation due to this “crazy project” is given on p.22 and the losses that will pertain to the natural environment are quantified on p.21.

Point 5: Since the Government is strong in implementing its plan, the alternatives to bring back balance in nature were also not discussed.

Response 5: If Canal Istanbul will be realized, the scope of the damage will be so high that we really do not think there will be alternatives to bring back balance in nature. TÜBÄ°TAK (Scientific and Technological Research Council of Turkey) as well as various studies by experts, including environmental engineers, marine biologists, geologists etc. state that risks are so enormous and damage will be irreparable. As the experts agree the unnatural merger of ecosystems of the Black Sea and the ​​Marmara Sea will completely change the whole ecosystem of both seas and will be a deadly blow especially for the Marmara Sea, which will have ramifications for the Aegean Sea as well. Also, we think that the struggle has not ended yet and the people of Istanbul, experts, professional chambers and NGO’s and various communities living in and around the area as well as the Istanbul Municipality still continue to resist this devastating project. At the same time, there are still court cases against the project. Therefore, we do not find it the right time to discuss remedial alternatives, as this will also carry the risk of being understood in an affirmative way. However, we discuss our stance on how to get out of this impasse on p. 28. paragraph 2.

Point 6: If the discussions are short and sharp, the paper will attract more readers and impact the ongoing development processes in Istanbul.

Response 6: We shortened the discussions. In order to do so, we cut down and removed repetitions, and moved some of the material to the previous section. We merged section 4. (Discussions) and section 5. (Conclusions).

Point 7: Repeating the same sentences may be avoided, and more graphics will enhance the succinctness.

Response 7: We eliminated repetitions and added new graphics in order to improve the manuscript (Figures 1, 2, 3, 9, 11, 14)

Reviewer 2 Report

Dear authors.

Thank you for your contribution, you did an amazing job, your title is attractive.

But for the current version of this manuscript, there are a lot of improvements that need to be applied. 

  1. I strongly recommend you use 'third person' to writing academic content, especially avoiding using the term 'we'.
  2. The abstract is up to 200 words, now your has 378.
  3. The abstract chapter should mainly be focusing on your research background, methods, results and conclusions. in your abstract, I can only see the background.
  4. For your case study, Canal Istanbul and Yenisehir Project, I strongly recommend you add a location map of these projects from the scope of the whole continent of Turkey. Not all of the readers are familiar with the geography of Turkey.
  5. This manuscript is more like a review instead of a research article. However, even as a review, the length of this manuscript is too long.
  6. The methodology of EIA and 1:00,000 MPR analysis are not enough, this manuscript just did a review for them.
  7. The conclusion chapter is more like an introduction. Without your research progress, also can write down the current conclusions.
  8. The format of the whole manuscript is chaotic, I downloaded and check the latest template of this journal, the current version of this manuscript did not have strict typesetting and format according to the requirements. Please revise the entire format thoroughly. e.g. Between line 117-118, there is a separation, but between line 181-182, there isn't. Between line 864-867, there are line 865 & 866 have no content, etc.
  9. There should be a full stop at the end of the sentence, line 1157, etc.
  10. What does the symbol at the end of the sentence mean, line 1211?
  11. ... ...

I hope the suggestions can benefit this manuscript.

Again, thank you for your contribution to Turkey's urban sustainable development.

Author Response

Response to Reviewer 2 Comments

Dear Reviewer,

Thank you so much for your constructive criticism. We improved our manuscript taking your suggestions into consideration. Below are the revisions made in response to your comments.

Once again, thank you very much for helping us to improve the manuscript. We hope that we have been able to respond to your suggestions adequately.

Best regards,

-------------------------------------------------------------------------------

Point 1: Are the research design, questions, hypotheses and methods clearly stated?

Response 1: We revised the abstract and added new paragraphs in Section 1. (Introduction) and Section 2. (Material and Methods) in order to clarify our research questions, and the material we used in our analysis. Our additions to the text are marked in YELLOW.

Point 2: Are the arguments and discussion of findings coherent, balanced and compelling?

Response 2: We eliminated repetitions, shortened where appropriate and added new phrases (marked in YELLOW) to enhance the clarity and coherence of arguments and discussions. We also did English language editing to improve and fine-tune the text.

Point 3: Abstract - strongly recommend you use 'third person' to writing academic content, especially avoiding using the term 'we'.

Response 3: We used “third person” where appropriate and eliminated the term “we” as much as we can. However, this style of writing is also used in writing academic content and is preferred by some academics. This style of writing is also used in other articles in this special issue. So, we changed it where we can.

Point 4: Introduction - The abstract is up to 200 words, now your has 378.

Response 4: We shortened it to 232 words.

Point 5: The abstract chapter should mainly be focusing on your research background, methods, results and conclusions. in your abstract, I can only see the background.

Response 5: We revised the abstract and considerably shortened it leaving out most of the background. We revised it to include underlying research questions, methods, results and conclusions.

Point 6: For your case study, Canal Istanbul and Yenisehir Project, I strongly recommend you add a location map of these projects from the scope of the whole continent of Turkey. Not all of the readers are familiar with the geography of Turkey.

Response 6: We added a number of new maps and plans to explain the context and the project. We added both the 1980 Plan (Figure 1) and the 2009 Plan (Figure 2), which shows the continuity of the planning approach among the two plans in directing the growth of the city in the east-west direction and limiting growth towards the north. Figure 1 also shows the location of Istanbul within Turkey. Figure 3 shows the extent of the project area, which includes the Canal Istanbul and Yenişehir Project as well as the Istanbul Airport. Also, Figures 4 and 5 show the project area and help to understand some of the existing features of the area where the project is located. Figure 7 shows the proposed route of the canal. Figure 9, again, shows the vast extent of the project area and the surrounding agricultural lands and forests. Figure 8 shows the portion of the 2009 plan concerning the project area, which shows clearly the area to the north of the Sazlıdere Dam is protected. Figure 10 shows the revision made where the Sazlıdere dam no longer exists and all the protected areas are opened to residential development shown in yellow.

Point 7: This manuscript is more like a review instead of a research article. However, even as a review, the length of this manuscript is too long.

Response 7:

In terms of the number of words in the main text, our manuscript is not much longer than the articles in this special issue. However, it is longer in the number of pages since we have a lot of visual material and many footnotes. We value the footnotes because they document this complex issue and will provide invaluable material for other researchers who might like to study the case of Canal Istanbul and YeniÅŸehir Project and the legal and institutional changes pertinent to planning.

On the other hand, the journal does not specify any restrictions about the length or limit the number of words.

The subject matter is quite complex and making it meaningful to a foreign audience requires more explanations than usual. We eliminated some parts. However, in order to contextualize our investigation and arguments, we had to add some new writings as well. In the end, it has not been possible to shorten the manuscript significantly.

Point 8: The methodology of EIA and 1:00,000 MPR analysis are not enough, this manuscript just did a review for them.

Response 8: We compared the area for which the plan revision was made to the decisions of the 2009 plan and made a qualitative evaluation in terms of changes proposed by the plan revision on land-uses, densities, transportation, irreversible ecological damage, damages pertaining the cultural heritage, water basins, agricultural lands, and forests, etc. In our revision of the manuscript, we also added quantitative information demonstrating the extent of the likely damage if the project is realized as proposed in the revision plan.

On the other hand, the EIA report is an extensive document of 1635 pages which covers a wide array of issues ranging from Policy and Institutional Framework; Analysis of Project Location, Technical Details of the Project, Construction and Operation Impacts, Socio-Economic, Environmental Impacts, Participation and so on. We limited our analysis to planning issues and principles regarding sustainability (please, see explanations on p.4, lines 186-191) and p.16, lines 545-564 of the revised manuscript)

Point 9: The conclusion chapter is more like an introduction. Without your research progress, also can write down the current conclusions.

Response 9: We revised the discussion and conclusion sections

Point 10: The format of the whole manuscript is chaotic, I downloaded and check the latest template of this journal, the current version of this manuscript did not have strict typesetting and format according to the requirements. Please revise the entire format thoroughly. e.g. Between lines 117-118, there is a separation, but between lines 181-182, there isn't. Between lines 864-867, there are lines 865 & 866 have no content, etc 

Response 10: We are really sorry for that. The template automatically formats the text. At least, that's what we thought. We paid utmost attention to it this time. There are still some format errors we would like to correct but the template does not allow us to do so. For instance, there are some extra spaces at the bottom of some pages. We try to eliminate them, but the template does not allow it. The visual material might also have an effect on this. If our manuscript gets accepted, we believe that we can handle these technical issues with the help of the publisher or someone professional who can help with it.

Point 10: There should be a full stop at the end of the sentence, line 1157, etc.

Response 10: Thanks a lot. Corrected.

Point 11: What does the symbol at the end of the sentence mean, line 1211? 

Response 11: Error. Erased.

Reviewer 3 Report

The paper, based on the topic, sought to understand Political dilemmas in the making of a sustainable city-region using Istanbul as the case study. The results presented are very interesting and relevant to the special issue of the journal. This notwithstanding, I do have a few comments for suggestions that I think could improve the manuscript prior to its acceptance and publication. Thus I recommend the paper be substantially revised and resubmitted for publication consideration. Please let me elaborate in my comments below.

Abstract

The abstract, although nicely written, could be improved by adding a clear problem statement, underlying research question/s, methodology used, the novelty of the study. This will help the readers to understand the scope of the paper.

Introduction

The Introduction is generally very well written. The arguments are well communicated with very good flow and clarity in the write up. On Lines 72 and 73 however, the authors stated/concluded that “In this context, market-based urban policies, which prioritized competitiveness and economic growth, endangered the sustainability and resilience of city-regions”. The preceding statements did not however clear explain why this conclusion is warranted. Probably a statement or two can be added to make this conclusion clearer.

On Line 75 and 76, “… increasingly became a priority imposed by the hegemonic power relations in a neoliberal context. And Istanbul is no exception”. The last statement should be revised, i.e., “And” should not begin the sentence.

Materials and methods

Line 168: Considering this to be an academic write up, I suggest that the phrase “shed light” be replaced with a more appropriate phrase like “explore, explain, better understand, etc..”

Line 197-200: “The research method is based on desk research and policy analysis of relevant legislation and reports of public institutions. The analyses of the technical reports prepared by the local government and professional chambers on the subject are evaluated in the article”. The authors should state/indicate the specific policies and technical reports reviewed and analysed to help readers appreciate what is presented in the Results section.

Results and discussion

  • The results presented are very comprehensive, informative and relevant to the scope of the special issue. They respond to the objectives/themes set out to be achieved under Section 2: 1) legal regulations and norms, 2) ecology, environmental values and cultural assets and 3) planning principles and public interest.
  • However, Section 4 (Discussion) appears to be a repetition of several sections of the Results. In the Results section, I see the authors to have already discussed the findings/results; hence, no need for Section 4. The authors can either (i) present just the results without the discussions (situating the findings in literature) and do a proper Discussion in a different section; or 2) Merge Sections 3 and 4, where appropriate. Doing this will make the paper more focused, minimise the repetitions and considerably cut down the length of the paper

Conclusions

The Section presents a summary of the major issues/results presented in the paper. I do however suggest that the authors suggest some recommendations to the issues presented and also suggest areas for future research.

Author Response

Response to Reviewer 3 Comments

Dear Reviewer,

Thank you so much for your constructive criticism. We improved our manuscript taking your suggestions into consideration. Below are the revisions made in response to your comments.

Once again, thank you very much for helping us to improve the manuscript. We hope that we have been able to respond to your suggestions adequately.

Best regards,

---------------------------------------------------------------------------------

Point 1: Are the research design, questions, hypotheses, and methods clearly stated?

Response 1: We revised the abstract and added new paragraphs in Section 1. (Introduction) and Section 2. (Material and Methods) in order to clarify our research questions, and the material we used in our analysis. Our additions to the text are marked in YELLOW.

Point 2: Are the arguments and discussion of findings coherent, balanced and compelling?

Response 2: We eliminated repetitions, shortened where appropriate, and added new phrases (marked in YELLOW) to enhance the clarity and coherence of arguments and discussions. We also did English language editing to improve and fine-tune the text.

Point 3: Abstract - The abstract, although nicely written, could be improved by adding a clear problem statement, underlying research question/s, methodology used, the novelty of the study. This will help the readers to understand the scope of the paper.

Response 3: We revised the abstract. One of the reviewers commented that it was too long and exceeded the word limit. Therefore, we had to considerably shorten the abstract leaving out most of the background, and revised it to include underlying research questions, methodology used, and the novelty of the study in making this striking example available to the English-speaking academic circles.

Point 4: Introduction - The Introduction is generally very well written. The arguments are well communicated with very good flow and clarity in the write-up. On Lines 72 and 73 however, the authors stated/concluded that “In this context, market-based urban policies, which prioritized competitiveness and economic growth, endangered the sustainability and resilience of city-regions”. The preceding statements did not however clear explain why this conclusion is warranted. Probably a statement or two can be added to make this conclusion clearer.

Response 4: Thank you for bringing this to our attention. We have not actually expressed what exactly we meant. We changed the sentence in order to better express what we actually meant by this statement as follows:

“In this context, profit-oriented, market-based urban policies, which prioritized competitiveness and economic growth at the expense of the environment and the social good, endangered the sustainability and resilience of city-regions.”

Point 5: On-Line 75 and 76, “... increasingly became a priority imposed by the hegemonic power relations in a neoliberal context. And Istanbul is no exception”. The last statement should be revised, i.e., “And” should not begin the sentence.

Response 5: Once again, thank you. We took out “And”

Point 6: Materials and Methods - Line 168: Considering this to be an academic write up, I suggest that the phrase “shed light” be replaced with a more appropriate phrase like “explore, explain, better understand, etc..”

Response 6: We replaced the word “shed light” with “explore,” and revised this sentence further to express our research question more clearly in which we set out to investigate how the state creates exceptionalities in order to provide an institutional setting in which the mega projects can be realized. We moved this sentence one paragraph up from Materials and Methods to Introduction and made it the last paragraph of Introduction

Point 7: Line 197-200: “The research method is based on desk research and policy analysis of relevant legislation and reports of public institutions. The analyses of the technical reports prepared by the local government and professional chambers on the subject are evaluated in the article”. The authors should state/indicate the specific policies and technical reports reviewed and analyzed to help readers appreciate what is presented in the Results section.

Response 7: We listed the important documents (laws, regulations and other legal documents, policy documents, and technical reports) we used in our analyses. They are added to paragraphs 3, 4, and 5 of the Materials and Methods section. The additions to the manuscript are marked in YELLOW.

Point 8: Results and discussion - The results presented are very comprehensive, informative, and relevant to the scope of the special issue. They respond to the objectives/themes set out to be achieved under Section 2: 1) legal regulations and norms, 2) ecology, environmental values, and cultural assets, and 3) planning principles and public interest.

However, Section 4 (Discussion) appears to be a repetition of several sections of the Results. In the Results section, I see the authors have already discussed the findings/results; hence, no need for Section 4. The authors can either (i) present just the results without the discussions (situating the findings in literature) and do a proper Discussion in a different section; or 2) Merge Sections 3 and 4, where appropriate. Doing this will make the paper more focused, minimize the repetitions, and considerably cut down the length of the paper

Response 8: Again, thank you very much for your input. We merged sections 3 and 4, where relevant and eliminated repetitions. We revised the discussion and included a shorter conclusion contextualizing our study.

Point 9: Conclusion - The Section presents a summary of the major issues/results presented in the paper. I do however suggest that the authors suggest some recommendations to the issues presented and also suggest areas for future research.

Response 9: We revised the conclusions and included our recommendations briefly. 

Round 2

Reviewer 2 Report

Dear authors,

Thank you for your revision, this manuscript improved a lot.

One minor reminder, normally, in a conclusion chapter, there shouldn't be a citation/reference. All of the content of the chapter conclusion should be your own words.

Well done!

Best

Author Response

Response to Reviewer 2 Comments

Dear Reviewer,

Once again, thank you so much for your constructive criticism, which gave us an opportunity to improve our manuscript taking. Below is our revision made in response to your comment on the 2nd round of review.

Best regards,

---------------------------------------------------------------------------------

Point 1: One minor reminder, normally, in a conclusion chapter, there shouldn't be a citation/reference. All of the content of the chapter conclusion should be your own words.

Response 1: We revised the last paragraph of our conclusion, which had citations, and reworded the ideas putting them into our own words. Our revision is marked in BLUE.

We moved the relevant citations to the 1st paragraph of 4.1., again marked in BLUE.